# Cerebellar involvement in an evidence-accumulation decision-making task

Ben Deverett[1,2,3]*, Sue Ann Koay[2], Marlies Oostland[2], Samuel S-H Wang[1,2]*

[1]Department of Molecular Biology, Princeton University, Princeton, United States; [2]Princeton Neuroscience Institute, Princeton University, Princeton, United States; [3]Rutgers Robert Wood Johnson Medical School, Piscataway, United States

**Abstract** To make successful evidence-based decisions, the brain must rapidly and accurately transform sensory inputs into specific goal-directed behaviors. Most experimental work on this subject has focused on forebrain mechanisms. Using a novel evidence-accumulation task for mice, we performed recording and perturbation studies of crus I of the lateral posterior cerebellum, which communicates bidirectionally with numerous forebrain regions. Cerebellar inactivation led to a reduction in the fraction of correct trials. Using two-photon fluorescence imaging of calcium, we found that Purkinje cell somatic activity contained choice/evidence-related information. Decision errors were represented by dendritic calcium spikes, which in other contexts are known to drive cerebellar plasticity. We propose that cerebellar circuitry may contribute to computations that support accurate performance in this perceptual decision-making task.
DOI: https://doi.org/10.7554/eLife.36781.001

## Introduction

Although the cerebellum is best known for its role in controlling movement, clinical and experimental evidence have long indicated that the posterior cerebellum regulates a wide range of cognitive functions (*Konarski et al., 2005*; *Schmahmann and Sherman, 1998*; *Stoodley et al., 2012*), including decision-making and working memory (*Blackwood et al., 2004*; *Desmond et al., 1997*; *Ernst et al., 2002*; *Kansal et al., 2017*). For example, focal cerebellar lesions in humans lead to impairment in working memory performance (*Gottwald, 2004*), and cerebellar fMRI activation increases with working memory demands (*Küper et al., 2016*). However, very little is known about the circuit mechanisms supporting these roles.

In the domain of movement control, the cerebellum is thought to use sensory and internal information as a means of adjusting action on a subsecond scale (*Krakauer and Shadmehr, 2006*). The cerebellar cortex consists of highly characteristic circuitry occurring in repeating modules which are likely to perform similar manipulations on information irrespective of whether the information is sensory, motor, or neither (*Popa et al., 2014*; *Reeber et al., 2013*). Thus, well-established models of cerebellar motor learning may be expanded to support the control of cognitive processing (*Ito, 2008*).

Neuronal correlates of the perceptual decision-making process have been studied using behavioral tasks in animal models (*Carandini and Churchland, 2013*) including evidence accumulation paradigms in which animals must continuously update the contents of working memory to guide a decision (*Brunton et al., 2013*; *Gold and Shadlen, 2007*; *Morcos and Harvey, 2016*; *Pinto et al., 2018a*). Behavioral performance in these tasks develops over time and can be marked by decision side biases, history effects, and error rates that diminish with training (*Pinto et al., 2018a*). The detailed mechanisms by which accurate decisions are formed and errors are reduced remain unsolved.

*For correspondence:
deverett@princeton.edu (BD);
sswang@princeton.edu (SS-HW)

Competing interests: The authors declare that no competing interests exist.

Neurons in multiple brain structures across species have been found to represent various stages in the transformation from sensory information to decision signals. These regions include prefrontal, premotor, parietal, and primary and secondary sensory cortices, striatum, midbrain structures, and possibly others (*Akrami et al., 2018*; *Brody and Hanks, 2016*; *Gold and Shadlen, 2007*; *Scott et al., 2017*; *Yartsev et al., 2018*). Many of these structures are reciprocally connected with the cerebellum, notably with posterior cerebellar regions such as crus I (*Kelly and Strick, 2003*; *Prevosto et al., 2010*; *Strick et al., 2009*). Communication between forebrain structures and the posterior cerebellum (*Buckner et al., 2011*; *Stoodley et al., 2017*) raises the possibility that the cerebellum might participate in the formation or updating of decision-related signals.

We investigated cerebellar neural activity during decision-making in a head-fixed rodent model. Like previously developed decision-making tasks (*Brunton et al., 2013*; *Morcos and Harvey, 2016*; *Shadlen and Newsome, 2001*), our task demands dynamic manipulation of working memory and decision-making under uncertainty, which recruit cerebellar activation in humans (*Blackwood et al., 2004*; *Kansal et al., 2017*), as well as the correction of errors, a cerebellar role that may extend beyond the motor domain (*Ito, 2008*).

## Results

### A decision-making task for cerebellar investigations

To study decision-making in the cerebellum, we developed a task with five key properties: (1) integration of evidence over seconds (*Scott et al., 2015*), (2) minimal movement until presentation of a readout cue (*Shadlen and Newsome, 2001*), (3) task structure to match established decision-making frameworks (*Brunton et al., 2013*), (4) sensorimotor engagement of the lateral posterior cerebellum (*Manni and Petrosini, 2004*), and (5) amenability to head-fixed conditions to facilitate two-photon imaging (*Dombeck et al., 2007*). In our evidence accumulation task (*Figure 1A*, *Video 1*), each trial contains a 3.8 s cue period in which a series of air puffs (pieces of evidence) is delivered to the left and right whiskers. Then, following a short delay period with no stimuli, lick ports are brought into the animal's reach and mice lick leftward or rightward to indicate which side received the greater number of stimuli, with a correct response leading to a water reward to end the trial. Puffs are generated randomly with differing rates on each side, demanding that mice continually attend to the stimuli to achieve optimal performance (*Brunton et al., 2013*).

Mice learned to perform this task with high accuracy (*Figure 1B*, *Figure 1—figure supplement 1*). Behavioral regression analysis demonstrates that mice used evidence throughout the entire cue period to guide decisions, with a bias for evidence toward the end of the cue period (*Figure 1C*), similar to some recency strategies that have been documented in human evidence accumulation (*de Lange et al., 2010*). Like other tasks in which movement is minimal until a go cue is presented (*Scott et al., 2017*; *Shadlen and Newsome, 2001*), mice learned not to lick during evidence presentation (*Figure 1E*). This task can therefore be used to study working memory, evidence accumulation, and decision-making under head-fixed conditions.

We focused our study on the ansiform area (crus I) (*Luo et al., 2017*) of the posterior hemispheric cerebellum (*Figure 1F*), a region that evolutionarily expanded in tandem with prefrontal cortex (*Balsters et al., 2010*) and communicates bidirectionally with forebrain regions including prefrontal, parietal, and somatosensory cortex (*Kelly and Strick, 2003*; *Prevosto et al., 2010*; *Proville et al., 2014*). This cerebellar region represents orofacial features under anesthesia (*Manni and Petrosini, 2004*; *Shambes et al., 1978*), suggesting that it might aid in processing complex task-related information. First, to determine whether activity in this cerebellar region participates in the decision-making behavior, we injected the GABA$_A$ agonist muscimol bilaterally into crus I. Inactivations reduced choice accuracy while leaving intact the ability to lick and perform trials (*Figure 1D*, *Figure 1—figure supplement 2*). To quantify the behavioral effects of the perturbation, we fit the inactivation data to a logistic regression model that considers the animal's choice on a trial-by-trial basis as a function of current evidence, the previous trial choice and outcome, and a bias (*Busse et al., 2011*; *Licata et al., 2017*). Fits to this model suggest that inactivations altered multiple behavioral parameters, notably including a reduction in animals' sensitivity to evidence and an increased tendency to make the same choice as in the previous trial (*Figure 1—figure supplement 2C*). Therefore, activity

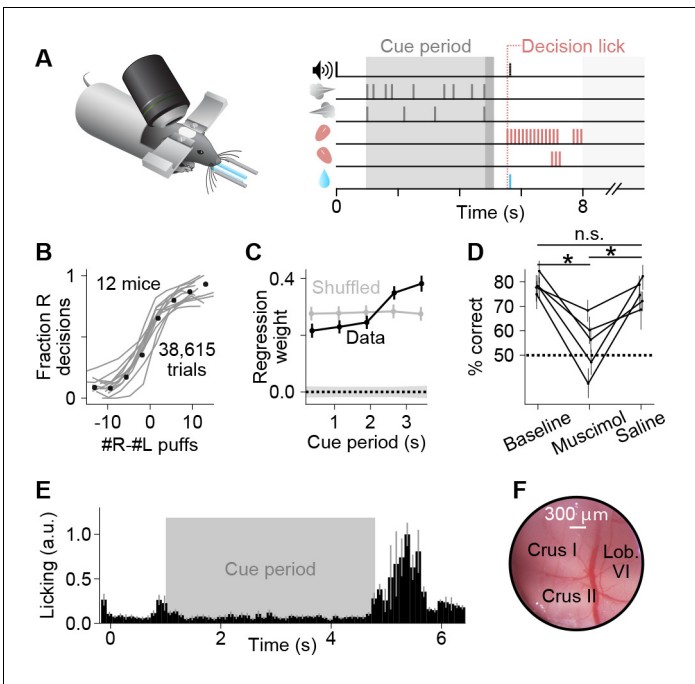

**Figure 1.** A somatosensory decision-making task that depends on the cerebellum. (**A**) In each trial, two streams of random, temporally Poisson-distributed air puffs were delivered to the left and right whiskers. After a delay, mice licked one of two lick ports indicating the side with more cumulative puffs to receive a water reward. Gray-shaded regions from left to right: cue period, delay, intertrial interval. Decision lick: first detected lick after the delay. (**B**) Psychometric performance data on the evidence accumulation task. Gray lines, individual mice; black points, average across all trials from all animals (n = 38,615 trials, 12 mice). (**C**) Logistic regression analysis correlating animal choice with cues delivered at different time bins of evidence presentation, demonstrating that the entire cue period was used to guide decisions. Each point indicates the magnitude of that time bin's influence on decisions (all points significantly greater than zero, Wald test, p<0.0001). For comparison, bins (gray points) or choices (shaded 1 s.d. gray zone) were shuffled. Error bars: 95% confidence interval. (**D**) Behavioral effect of bilateral injections of muscimol or saline into crus I, compared to baseline performance with no injections. Each set of joined points represents one mouse. Error bars: 95% confidence interval. *p<0.05, n.s.: not significant (two-tailed paired t-test). (**E**) Movie-based licking measurements from mice over the duration of trials. Bar heights show mean ±s.e.m. across animals of trial-averaged licking signals. (**F**) Example cranial window over the left posterior hemispheric cerebellum, indicating the site of imaging and inactivation.

DOI: https://doi.org/10.7554/eLife.36781.002

The following figure supplements are available for figure 1:

**Figure supplement 1.** Behavioral performance in the decision-making task.

DOI: https://doi.org/10.7554/eLife.36781.003

**Figure supplement 2.** Behavior in inactivation experiments.

DOI: https://doi.org/10.7554/eLife.36781.004

**Figure supplement 3.** Muscimol injection sites.

DOI: https://doi.org/10.7554/eLife.36781.005

in this region is necessary for successful performance of the task, suggesting it may play a role in decision-making computations.

## Purkinje cell somatic calcium encodes task-relevant information

In previously investigated brain regions, neurons exhibit choice- and evidence-specific modulations of activity over the duration of evidence accumulation and decision-making (*Ding and Gold, 2012*; *Hanks et al., 2015*; *Latimer et al., 2015*; *Scott et al., 2017*; *Shadlen and Newsome, 2001*). To test for choice- and evidence-related activity in Purkinje cells, we imaged somatic calcium, which follows modulations in simple-spike rate (*Lev-Ram et al., 1992*; *Ramirez and Stell, 2016*), using the genetically encodable calcium indicator GCaMP6f in mice performing the decision-making task (*Figure 2A,*

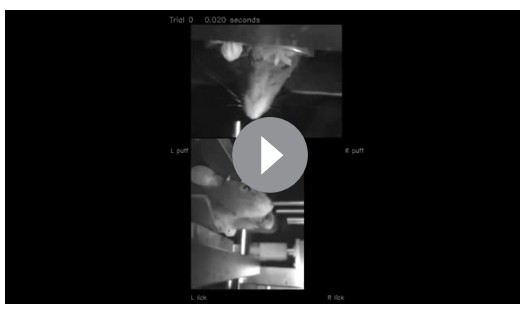

**Video 1.** Example trials of a mouse performing somatosensory evidence accumulation. Flashes along the sides indicate air puffs delivered to the whiskers. Flashes along the bottom indicate detected licks.
DOI: https://doi.org/10.7554/eLife.36781.006

*B*). We imaged a total of 843 Purkinje cell somata in four mice. We found a population of cells in which calcium activity was modulated during the cue period, exhibiting increases or decreases in fluorescence spanning the duration of evidence accumulation and decision formation (*Figure 2C–E*). In 70% of cells, cue-period fluorescence was better correlated with time than pre-cue-period fluorescence was (95% CI: 67–72%, bootstrap). This was significant compared to when cue and pre-cue period identity was shuffled (49% of cells; 95% CI: 46–52%). These modulations were sometimes evident at the level of individual trials (*Figure 2—figure supplement 1*). At the end of each trial, activity returned to baseline (p=0.91, two-tailed paired t-test).

Cytoplasmic calcium acts as a temporally filtered readout of firing rate, and calcium extrusion in Purkinje cells occurs on a slower time scale (see *Figure 3B*; *Konnerth et al., 1992*; *Lev-Ram et al., 1992*; *Fierro and Llano, 1996*; *Rokni and Yarom, 2009*; *Ramirez and Stell, 2016*) than in neocortical neurons (*Chen et al., 2013*). Therefore, our observed increasing and decreasing time courses of calcium could reflect various firing rate profiles, such as impulse responses, ramps, or steps. We did find that electrically recorded Purkinje cells exhibited gradually increasing rates of firing throughout the cue period (*Figure 2—figure supplement 2*).

Elsewhere in the brain, neuronal activity during evidence accumulation and decision-making can encode behavioral variables of choice and evidence (*Gold and Shadlen, 2007*; *Latimer et al., 2015*; *Scott et al., 2017*). To determine whether predictive behavioral information was represented in the population activity of these Purkinje cells, we constructed linear classifiers based on all somatic signals in each animal. These classifiers accurately decoded the upcoming choice and the side with greater evidence (*Figure 2F*), indicating that as a population, the imaged neurons encode behaviorally relevant features of the decision-making process.

Because choice and evidence are correlated when mice successfully perform the task, we asked whether choice- or evidence-related information existed independently at the population level in the neuronal signals. To separate the two, we determined how decoding accuracy changed after removing information about one of the variables, by shuffling its identity across trials while holding the other variable constant. For example, when the choice on each trial was randomly assigned to another trial with the same sensory evidence, choice decoding accuracy dropped significantly (*Figure 2F*, top panel, top two traces). The difference in decoding accuracy between the original and shuffled data indicates the magnitude of independent choice-related information in the population-level neural activity. We performed the converse test as well, shuffling evidence while holding choice constant, and found that evidence-related information is also represented independently in population-level neuronal activity (*Figure 2F*, bottom panel). Therefore, somatic signals encode both choice- and evidence-related information.

The encoding of choice and evidence variables suggests that these neurons might play a role in decision-making computations. However, in an alternative hypothesis, the somatic signals we observed might represent motor behaviors that occur as an independent consequence of the decision-making process, for example by encoding motor commands for licking or other movements. The imaged region is known to encode primarily orofacial features in rodents (*Bosman et al., 2010*; *Manni and Petrosini, 2004*). To test for pre-decision movements, we used camera recordings to measure licking as well as five other motor behaviors during evidence accumulation for trials with differing evidence and choices (*Figure 2—figure supplements 3* and *4*, *Videos 2* and *3*). Licking, nose, whisker, and forepaw movements did not differ across trial types and were unable to predict choice and evidence variables. Therefore, Purkinje cells encode choice- and evidence-related variables with minimal information about predictive anticipatory movements.

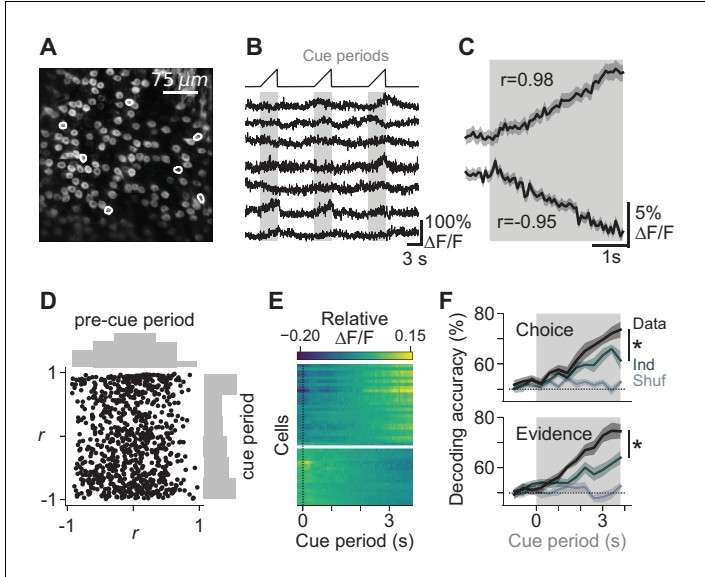

**Figure 2.** Task-dependent modulation of Purkinje cell somatic calcium signals. (A) Example two-photon field of view of Purkinje cell somata. (B) Traces of extracted calcium signals from somata indicated in (A). Shaded regions and ramps at top indicate cue periods. (C) Trial-averaged activity during evidence presentation from two example cells. Modulation index *r* was defined as the Pearson correlation between the averaged signal and time in the cue period. Confidence interval on traces indicates s.e.m. (D) Cue-period fluorescence modulation in all imaged somata (n = 4 mice, 843 cells). Modulation index *r* was computed preceding the cue period ('pre-cue') and during the cue period. (E) Trial-averaged activity during the cue period of neurons with the highest absolute modulation index (top 5%) in each session. ΔF/F signals are mean-subtracted. (F) Output of a linear decoder predicting the animal's upcoming choice and the side with more evidence on a trial-by-trial basis using somatic data from the cue period of each trial. Each trace represents the mean ±s.e.m. (n = 6 sessions in four mice). Choice: side of the animal's decision. Evidence: side with more evidence. Gray-shaded regions: cue period. Shuffle: relevant variable (choice or evidence, respectively) was shuffled across trials. Ind: relevant variable (choice or evidence, respectively) was shuffled while holding the other variable constant, to compute the independence of encoding of the relevant feature. *: p<0.01 (paired t-test using cue-period-only data).

DOI: https://doi.org/10.7554/eLife.36781.007

The following figure supplements are available for figure 2:

**Figure supplement 1.** Somatic signals are modulated on individual trials.
DOI: https://doi.org/10.7554/eLife.36781.008

**Figure supplement 2.** Electrical recordings from Purkinje cells during behavior.
DOI: https://doi.org/10.7554/eLife.36781.009

**Figure supplement 3.** Movie-based licking measurements.
DOI: https://doi.org/10.7554/eLife.36781.010

**Figure supplement 4.** Movements do not explain somatic signals.
DOI: https://doi.org/10.7554/eLife.36781.011

## Dynamics of choice- and evidence-related information in Purkinje cells

To determine how individual Purkinje cells represented choice, we examined their coding properties in left- and right-choice trials. In 80% (678/843) of cells, cue-period calcium was modulated in the same direction (i.e. upward or downward) without regard to whether the upcoming decision was left or right, while in the remaining 20% (165/843) of cells, activity for left choices and right choices was modulated in opposite directions (*Figure 3A*). In 30% (256/843) of cells, pre-decision fluorescence (measured in the 500 ms preceding the end of the delay) differed significantly between L-choice and R-choice trials (criterion p<0.05, two-tailed t-test). Of these choice-selective cells, 63% (162/256) exhibited greater activity in left-choice trials, compared with 37% (94/256) in right-choice trials. While recordings from a single (left) hemisphere might have been expected to produce strongly lateralized representations, these mixed representations of left and right choices are consistent with neocortical recordings in decision-making, particularly in frontal regions (*Erlich et al., 2011*).

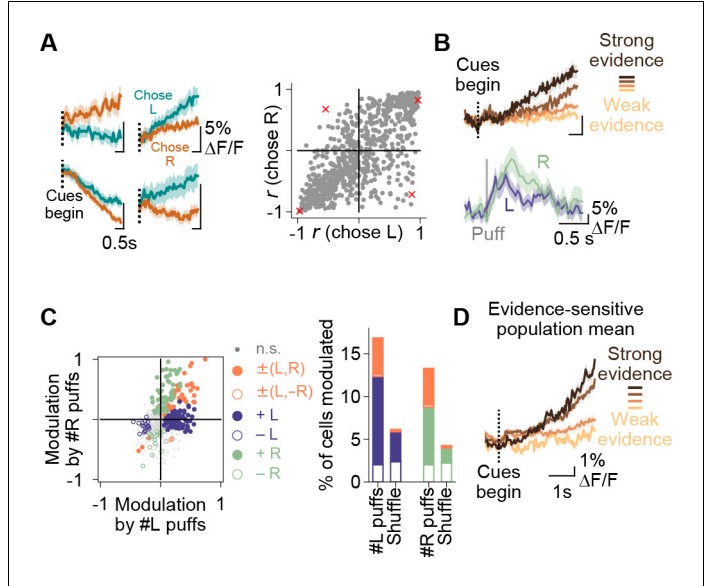

**Figure 3.** Purkinje cell representations of choice and evidence. (**A**) Left: mean activity of four example somata during the cue period, split according to the choice made in each trial. Traces represent mean ±s.e.m. over all trials of a particular choice. Right: summary of the relationship between modulation index $r$ and animal choice for all imaged cells. Red x's: cells shown on left. (**B**) Top: mean cue-period activity in correct trials from one example cell, split according to the strength of evidence presented (strong: #L puffs > 9; weak: #L puffs < 2). Bottom: mean puff-triggered response of one example cell to left (L)- and right (R)-sided puffs. Mean $t_{1/2\ decay}$: 406 ms. Shading: s.e.m. (**C**) A linear model was used to determine the influence of left- and right-sided puffs on pre-decision fluorescence activity for each cell over all trials. Left: each dot represents one cell. Modulation: normalized coefficient of the linear fit between puff number and fluorescence. Colored data points indicate cells with significant coefficients. Right: Proportion of cells in each category on left. Shuffle: puff counts were shuffled across trials of the same choice before regression. Percent of modulated cells is significantly above the shuffle for the +L, +R and ±(L,R) conditions (p<0.0001, two-tailed z-test). (**D**) Mean cue-period activity in correct trials across all evidence-modulated cells, split according the level of evidence presented in the trial (strong: #pref side puffs-#nonpref side puffs > 8; weak: #pref side puffs-#nonpref side puffs<-8).

DOI: https://doi.org/10.7554/eLife.36781.012

The following figure supplement is available for figure 3:

**Figure supplement 1.** Somatic modulations are absent in a task-free context.
DOI: https://doi.org/10.7554/eLife.36781.013

We next asked how Purkinje cells represented evidence. We observed cells in which cue-period activity was modulated by the strength of

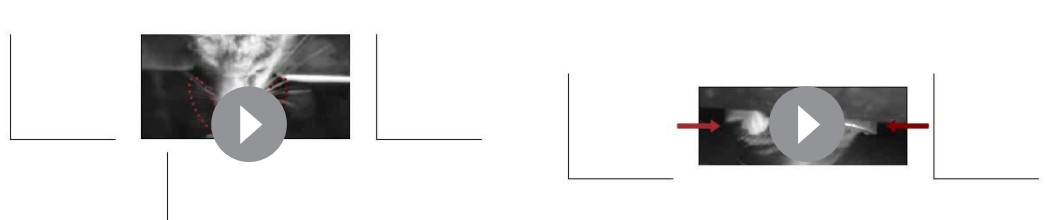

**Video 2.** Measurement of orofacial movements from behavioral movies. Traces represent the extracted movement metric (see Methods) from the corresponding regions outlined in the movie.
DOI: https://doi.org/10.7554/eLife.36781.014

**Video 3.** Measurement of forepaw movements from behavioral movies. Traces represent the extracted movement metric (see Materials and methods) from the denoted paws.
DOI: https://doi.org/10.7554/eLife.36781.015

evidence presented, and responses to individual sensory events were apparent in some cells as puff-triggered averages that rose and fell in approximately 1 s (*Figure 3B*). Therefore, to quantify the extent to which the strength of evidence affected the activity of each neuron, we used linear regression to fit trial-by-trial dependence of pre-decision fluorescence on evidence quantity (*Figure 3C*), where evidence was defined as the total number of right puffs (#R) or left puffs (#L) in a trial. Based on the significance and coefficients of these fits, each cell was categorized as having either a positive (+) or negative (-) relationship between fluorescence and evidence on the left (#L), right (#R), or both (#L,#R) sides. We found significant relationships in 26% (216/843) of neurons, with most cells exhibiting a correlation with single-sided evidence (+L, -L, +R, and -R; 178 cells), and a smaller number showing a correlation with a linear sum or difference of evidence (±(L,R)/±(L, -R), 38 cells). Therefore, individual cells were predominantly but not exclusively sensitive to evidence presented on one side, consistent with properties of some neocortical neurons in evidence accumulation (*Scott et al., 2017*). As a population, these evidence-modulated cells encoded the strength of evidence presented for decision-making (*Figure 3D*). In animals not performing the decision-making task, cue-period fluorescence modulation, evidence side decoding, and evidence modulation were not observed (*Figure 3—figure supplement 1*), indicating that the signals we observed are task-specific and are not consequences of baseline Purkinje cell response properties. The evidence-related representations we observed do not demonstrate precise moment-to-moment integration of evidence that is thought to occur in some forebrain neurons (*Gold and Shadlen, 2007*; *Hanks et al., 2015*), but they do suggest an engagement of the cerebellum in the processing of important task variables.

## Error-associated signaling in Purkinje cell dendrites

In theories of cerebellar learning, the transformation of mossy-fiber input to Purkinje cell output is refined by climbing fiber-driven error signals which drive plasticity (*Marr, 1969*). These instructive error signals evoke calcium transients in Purkinje cell dendrites (*Ozden et al., 2009*; *Tank et al., 1988*) and an accompanying complex electrophysiological spike (*Llinás and Sugimori, 1980*). To test for task-related activity in this pathway, we imaged calcium using GCaMP6f in Purkinje cell dendrites (*Figure 4A,B*). We observed that in many cells, dendritic events occurred directly following the animal's decision, specifically when that decision was an error (*Figure 4C–E*). In 82% of cells, the mean activity following errors exceeded that following rewards (p<0.0001, Wilcoxon signed-rank test). This increase in activity occurred in both left- and right-choice trials, in which sensory events differed, suggesting that the signal was reporting a task feature that was independent of pre-decision evidence.

When mice make errors, their behaviors, especially their licking patterns, differ from correct trials. We therefore tested whether the elevation in dendritic signaling could reflect a purely motor event such as a lick-cessation signal, since mice cease licking at moments of error (*Figure 4—figure supplement 1*). Such a lick-cessation signal should occur not just at moments of error, but whenever licking ceases, including in correct trials. We therefore measured dendritic signals at every instance of lick cessation in both error and correct trials, and compared the magnitude of the signals across the two contexts. We found that lick-cessation-aligned dendritic signalling was significantly elevated in error trials relative to correct trials (*Figure 4F*), indicating that our results are not explained by lick-cessation signals. We tested a number of similar hypotheses, including lick initiation events, orofacial movements, varying licking magnitudes, and trials in the absence of auditory cues, and found that dendritic signals were also significantly error-modulated in all cases (*Figure 4F*, *Figure 4—figure supplement 1*). Thus dendritic events encode an error-associated signal that is not specific to measured parameters of movement.

Dendritic signalling was consistently elevated in error trials relative to correct trials across varying trial difficulties, with a modest but non-significant reduction in magnitude in trials with stronger evidence (*Figure 4—figure supplement 1*). These error-associated events may potentially represent a training signal which can be useful to guide learning (*Schultz et al., 1997*). Indeed, we found that the population of Purkinje cells could decode trial outcome (correct or error) on a trial-by-trial basis (*Figure 4G*, post-choice correct/error decoding greater than shuffle and pre-choice conditions, p<0.01, two-tailed paired t-test).

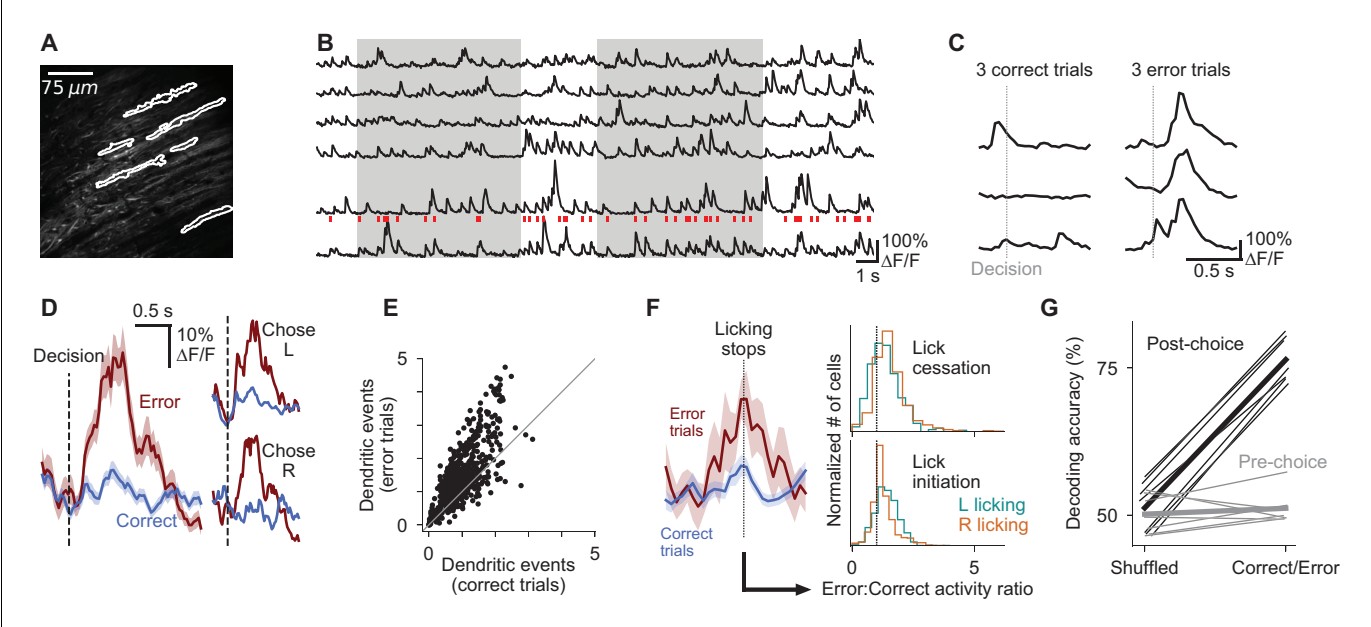

**Figure 4.** Purkinje cell dendrites encode decision errors. (A) Example two-photon field of view of Purkinje cell dendrites. (B) Signals extracted from cells indicated in (A). Red ticks: dendritic calcium transients extracted from the bottom trace. (C) Activity of one cell in six trials, aligned to the moment of the decision lick. (D) Mean activity of one example cell aligned to the moment of the decision lick. Left: activity is divided into correct and error trials. Right: activity is further divided into left-choice and right-choice trials. Error shading indicates s.e.m. (E) Summary of mean activity in the 800 ms following reward delivery (correct trials) or lack thereof (error trials) (n = 6 mice, 599 cells). (F) Left: mean response of an example dendritic signal aligned to moments when licking ceased, split according the outcome of the trial in which the lick cessation occurred. Right: histograms indicating the magnitude of dendritic activity measured at moments when animals ceased (top) or initiated (bottom) licking, presented as a ratio of activity in error vs correct trials; cells with values greater than one exhibited increased activity when lick-cessation/initiation events occurred with errors, in comparison to the same motor event in correct trials. Error activity is elevated in a significant fraction of cells for all four histograms shown (p<0.0001, Wilcoxon signed-rank test). (G) Outcome (correct/error) decoding on a trial-by-trial basis using neuronal population activity in the period following reward delivery or lack thereof (post-choice), or the period preceding the decision (pre-choice). One line per behavioral session (n = 7 sessions, six mice). Thick lines: mean across sessions.

DOI: https://doi.org/10.7554/eLife.36781.016

The following figure supplement is available for figure 4:

**Figure supplement 1.** Error-associated behaviors do not explain the error-associated dendritic response.
DOI: https://doi.org/10.7554/eLife.36781.017

## Discussion

The present work reports the necessity of the cerebellum in an evidence-accumulation-based decision-making task. We have identified two Purkinje-cell signals that may contribute to this process: somatic activity that reflects evidence and choice, and dendritic signals that report errors. This convergence of task-relevant information onto Purkinje cells suggests that cerebellar activity may play important roles in decision-making, consistent with established hypotheses of cerebellar function in complex domains (*Ito, 2008*).

Cerebellar crus I communicates with numerous forebrain structures including somatosensory, frontal, and parietal regions (*Prevosto et al., 2010*; *Proville et al., 2014*; *Strick et al., 2009*) via the ventral dentate nucleus (*Bernard et al., 2014*; *Parker et al., 2017*) and thalamic intermediates (*Asanuma et al., 1983*; *Dum and Strick, 2003*). Posterior hemispheric cerebellar cortex and its principal target, the dentate nucleus, show sensorimotor activity relating to whisker sensation (*Bosman et al., 2010*), licking (*Gaffield et al., 2016*), and reward (*Wagner et al., 2017*), as well as preparatory activity (*Middleton and Strick, 1998*; *Popa et al., 2017*) and firing rate ramps (*Ashmore and Sommer, 2013*) that can influence thalamocortical circuits (*Parker et al., 2017*). The cerebellum is thought to use information from elsewhere in the brain to form internal models that predict and modulate brain activity (*Ito, 2008*; *Marr, 1969*; *Wolpert et al., 1998*). In the context of

our decision-making task, the lateral posterior cerebellum may receive evidence/decision-related efference copy from forebrain regions, where evidence and decision-related variables have been observed and proposed to support decision-making (*Ding and Gold, 2012*; *Hanks et al., 2015*; *Licata et al., 2017*; *Morcos and Harvey, 2016*; *Shadlen and Newsome, 2001*). Thus, the cerebellum is positioned to be part of a closed feedback-loop circuit in which it both receives and sends task-related information.

Previous studies have established a sophisticated conceptual framework for understanding the computational basis for evidence accumulation and decision-making (*Brunton et al., 2013*; *Gold and Shadlen, 2007*; *Juavinett et al., 2018*; *Morcos and Harvey, 2016*; *Scott et al., 2017*). Our results are a first stage of discovery suggesting that the cerebellum may constitute an additional node in the distributed network of regions that support this process (*Pinto et al., 2018b*). Muscimol disrupted the proportion of correct choices without disrupting the ability to make a choice. This suggests that the lateral posterior cerebellum modulates not the mechanics of action, but rather processes that precede the brain's commitment to act. Our fits to a behavioral choice model indicate that reduced performance was accompanied by a decreased weighting of evidence and increased weighting of choice history parameters. The increased dependence on trial history is interesting in light of recently reported sensory history effects in parietal cortex (*Akrami et al., 2018*). Complementary to association areas in the neocortex, signals emerging from the cerebellum are known to reach thalamic targets which send widespread projections throughout the brain (*Strick et al., 2009*), situating the cerebellum in a position to modulate one or many components of forebrain processing.

Our imaging of Purkinje cell somata revealed ramps of fluorescence. Cytoplasmic calcium acts as a temporally filtered readout of firing rate, limited by calcium removal times that are slower in Purkinje cells (see *Figure 3B*; *Konnerth et al., 1992*; *Lev-Ram et al., 1992*; *Fierro and Llano, 1996*; *Rokni and Yarom, 2009*; *Ramirez and Stell, 2016*) than in neocortical neurons (*Chen et al., 2013*). Preliminary electrical recordings also showed ramps, consistent with the idea that temporally filtered firing rate ramps may account for the observed fluorescence signals.

These somatic signals represented task-relevant information related to choice and evidence variables, although it remains an open question as to whether these signals precisely track accumulated evidence over time. They could exhibit firing ramps (*Shadlen and Newsome, 2001*), steps (*Latimer et al., 2015*), or more complex response profiles that form a temporal basis for evidence accumulation (*Scott et al., 2017*). The evidence representations we observed were primarily of single-sided evidence, consistent with neural recordings in PPC and FOF of rats performing a similar task (*Scott et al., 2017*). This could indicate that cerebellar involvement is upstream of the calculation of the decision variable (#R-#L), and we consider this a likely possibility. It is also notable that some studies (*Scott et al., 2017*; *Scott et al., 2015*) have suggested that the decision-making process may be supported by two weakly coupled single-sided accumulators, which may modify the interpretation of our results. Whichever the case may be, the neural activity we observed contains task-relevant information that may be used during evidence accumulation and decision-making.

Cerebellar theories propose that the mossy fiber-granule cell pathway encodes contextual or efference-copy signals which are used to generate short-term predictions (*Shadmehr et al., 2010*). Climbing fiber activity may shape the processing of granule cell inputs by inducing plasticity at multiple cerebellar sites (*Albus, 1971*; *Marr, 1969*; *Medina and Lisberger, 2008*). For example, climbing fiber-derived error signals may modify synaptic weights at parallel fiber-Purkinje cell synapses, providing a mechanism for weighting the contextual signals entering the cerebellum. This motivated us to ask whether in this task, signals may be observed in this pathway that report errors, for example of the outcome of the animal's choice. We observed an excess of dendritic calcium events coincident with decision errors, which has not been previously reported in the cerebellum. If the error-associated response were analogous to dopamine reward prediction errors, one might have expected strong modulation of the error response magnitude by trial difficulty, with the easiest trials producing the largest error response. However, no such trend was apparent.

We suppose that these dendritic signals might not represent graded information but rather a more categorical signal for updating the cerebellar representation. It might alternatively be the case that the slight but non-significant trend we did observe, which appears inverted relative to traditional reward error signals, could be an example of inverted reward signalling seen elsewhere in the brain (*Cohen, 2007*; *Matsumoto and Hikosaka, 2007*). In all cases, the consequences of this error signalling could be reflected in the behavioral learning of the animal, as found via trial-by-trial

analyses in some motor tasks (*Brooks et al., 2015*; *Medina and Lisberger, 2008*; *Ten Brinke et al., 2017*), but such effects are difficult to resolve in decision-making tasks like ours where learning is slow, spanning a period of many days or weeks.

The involvement of the cerebellum, with its clearly delineated cell types and connectivity (*Dean et al., 2010*; *Ito, 2012*) opens many attractive avenues for future studies in decision-making. The data presented in this study suggest multiple possible roles for cerebellar involvement in evidence-accumulation-based decision-making. For example, output signals from the cerebellum may be combined with signals in sensory circuits to control the input gain of sensory information into accumulators elsewhere in the brain. This model would be consistent with observations of cerebellar involvement in gating sensory information (*Apps et al., 1997*; *Ozden et al., 2012*) and inputs to working memory (*Baier et al., 2014*; *Sobczak-Edmans et al., 2016*). In a second possibility, the cerebellum may modulate dynamics of the accumulation process. Finally, cerebellar signals may modulate activity that converts the accumulator value into a decision. Such a post-categorization influence has been observed in prefrontal regions during evidence accumulation (*Erlich et al., 2015*). Detailed inactivation studies with high spatial and temporal precision can resolve these alternatives. In all cases, activity from the cerebellum may be combined with activity in forebrain structures to produce a refined signal that is more likely to yield a reward.

## Materials and methods

### Mice

Experimental procedures were approved by the Princeton University Institutional Animal Care and Use Committee and performed in accordance with the animal welfare guidelines of the National Institutes of Health. Data for the behavioral task came from 12 mice (six female, six male, 8–9 weeks of age at the start of experiments) of genotypes *Pcp2*-Cre (five mice, *Pcp2*-Cre line derived from The Jackson Laboratory, Stock #010536, RRID:IMSR_JAX:010536) and *Pcp2*-Cre-Ai148 or Ai148 (seven mice, Ai148 line acquired from Hongkui Zeng, Allen Brain Institute); for Purkinje cell dendritic imaging from six mice (four male, two female; 5 *Pcp2*-Cre, 1 *Pcp2*-Cre-Ai148), for Purkinje cell somatic imaging from six mice (two female, four male; 3 *Pcp2*-Cre, 3 *Pcp2*-Cre-Ai148), for inactivation experiments from six separate mice (three female *Pcp2*-Cre-Ai148, two male *Pcp2*-Cre-Ai148, one female Ai148; one was used in behavioral data but was never subjected to inactivation), and for electrophysiology experiments from another three mice (three male *Pcp2*-Cre-Ai148). Mice were housed in a 12 hr:12 hr reverse light:dark cycle facility, and experiments were performed during the dark cycle. During the experimental day, mice were housed in darkness in an enrichment box containing bedding, houses, wheels (Bio-Serv Fast-Trac K3250/K3251), climbing chains, and play tubes. At other times, mice were housed in cages in the animal facility, in groups of 2–4 mice per cage. Mice received 1.0–1.4 mL of filtered water per day. Body weight and condition was monitored daily.

### Surgical procedures

Mice were anesthetized with isoflurane (5% for induction, 1.0–2.5% for maintenance) and underwent surgical procedures lasting 3–4 hr. For mice in imaging experiments, a 3-mm-diameter craniotomy was drilled over the left posterior hemispheric cerebellum. In *Pcp2*-Cre imaging mice, AAV1.CAG. Flex.GCaMP6f.WPRE.SV40 (Penn Vector Core) virus was injected into crus I and surrounding regions, 220–280 μm below the brain surface (two injections, 200 nL per injection at 20 nL/min), using borosilicate glass pipettes (World Precision Instruments, 1B100F-4, 1/0.58 mm OD/ID) beveled to 30 degrees with a ~ 10 μm tip opening, and an automated injector system (World Precision Instruments Micro4). In all imaged mice, a window composed of a cannula (Ziggy's Tubes and Wires, 316 S/S Hypo Tube 9R GA. 0.1470/0.1490' OD x 0.1150/0.1200' ID x 0.0197' long) glued (Norland Optical Adhesive 71) to a glass coverslip (Warner Instruments 64–0720) was cemented atop the craniotomy, then a custom-machined titanium headplate (*Dombeck et al., 2007*) was cemented to the skull using dental cement (C and B Metabond, Parkell Inc.). In *Pcp2*-Cre mice for inactivations, small (~375 μm radius) craniotomies were drilled over left and right crus I, and guide pedestals (Plastics One, C315GS-5/0-.4 Guide 26GA 5 mm pedestal, cut to 0 mm) were implanted over each. After surgery, dummy cannulas were kept in guide pedestals at all times when injections were not being performed, and were changed approximately every 3 days. For electrophysiology experiments,

headplates as described above were implanted, and a 2 mm craniotomy was drilled over crus I and the dura removed. Per animal, one 64-channel silicon probe (Neuronexus, Buzsaki64-H64LP_30 mm in two mice and A4 × 16-Poly2-5mm-23s-200–177 H64_30 mm in one mouse) was placed on a custom-printed probe holder (designed in Blender and printed on a Formlabs Form2 3D printer), and lowered into crus I. During lowering of the probe, recordings were performed as described below to determine the final location of the probe. Ground and reference wires were attached to two stainless steel screws (000–120 × 1/16 SL bind machine screws, Antrin miniature specialties, Inc) above the forebrain. Absolute Dentin (Parkell) was used to secure the probe to the skull, and to create a well which was filled with silicone gel (Dow Corning 3–4680) and ophthalmic ointment (Puralube) to cover the craniotomy before being further secured using dental cement (C and B Metabond, Parkell Inc.) and dental acrylic (Jet Denture Repair, Lang Dental Manufacturing Co.). All animals were given buprenorphine (0.1 mg/kg body weight) and rimadyl (5 mg/kg body weight) after surgery and were given at least 5 days of recovery in their home cages before the start of experiments.

## Equipment

Behavioral training, inactivation, and imaging experiments were performed in a light-proof and sound-dampened chamber, with white noise playing at all times. Mice were seated in a custom aluminium tube (9 cm long, 3.5 cm inner diameter) with their head protruding out the front and an opening on the top of the tube for access of the imaging objective. For two mice in one session each, the mice stood on a cylindrical treadmill instead the tube. Mice were head-fixed using custom head bars screwed into the head plate, angled with the anterior end downward at 18.5 degrees from the horizontal. A set of custom-machined polyoxymethylene alignment tools was used to align the headplate in the lateral axis and in yaw for precise repositioning across days.

Air puffs were produced by activation of solenoids (NResearch, standard two-way normally closed isolation valve, 161T011) with input from an air source regulated to 10 psi (ControlAir Type 850 Miniature Air Pressure Regulator). Air was delivered via two tubes (Ziggy's Tubes and Wires, 316 S/S Hypo Tube 16T GA. 0.0645/0.0655' OD x 0.0525/0.0545' ID) custom-machined with uniform openings, and positioned parallel to one another, parallel to the anteroposterior axis of the animal, 10 mm apart mediolaterally and ~1 mm anterior to the nose of the animal. Water rewards were produced via activation of similar solenoids, and delivered via similar tubes positioned 4.5 mm apart from one another mediolaterally, and 0.5–1.0 mm anterior to the opening of the animal's mouth. Licks were detected via completion of one of two parallel electrical circuits for the left and right ports. Puff ports and lick ports were positioned in individual custom-machined polyoxymethylene brackets. The lick port bracket was mounted to a linear actuator (Actuonix, L16/P16 Mini linear actuator with feedback) which enabled retraction of the ports to and from the reach of the animal's tongue (within approximately 300 ms). The ports, brackets, and actuator were mounted via a custom-machined bracket to a micromanipulator (Sutter Instrument Company, MP-285 motorized micromanipulator) for precise positioning of the apparatus in a unique position for each animal. The entire experimental apparatus was mounted on a translatable stage (Danaher Precision Systems) atop a rotating optical breadboard (Thorlabs, RBB12A) to allow custom positioning in the x, y, and rotational axes for imaging. Mice were positioned according to a set of unique coordinates for each mouse on the stage, micromanipulator, and alignment pieces, which were maintained across behavioral sessions.

Behavioral movies were acquired using two USB cameras (Playstation Eye), modified by removal of infrared filters and encasings. One camera was positioned directly below the animal's mouth and the other at the side of the animal's face. Images were acquired at 30 Hz with 320 × 240 pixel resolution. Illumination was provided by an infrared LED array (Yr.seasons 48-LED Illuminator Light CCTV 850 nm IR Infrared Night Vision). Sounds were delivered to the apparatus by a speaker (Sony Tweeter XS-H20S) mounted below the apparatus.

All behavioral equipment and data collection were controlled by custom multi-process software written in Python (https://github.com/wanglabprinceton/accumulating_puffs; *Deverett, 2018a*; copy archived at https://github.com/elifesciences-publications/accumulating_puffs). A DAQ board (National Instruments, NI PCI-MIO-16E-4) was used to deliver and read electrical signals from the experimental apparatus. Solenoids were controlled using digital outputs to custom transistor-based switch circuits. Electrical signals corresponding to licks, solenoid signals, and microscope galvanometer position were acquired using analog inputs at 500 Hz. Camera signals were acquired using a

custom Python wrapper to the CLEye API (Code Laboratories, https://codelaboratories.com/downloads). The linear actuator was controlled using an LAC board (Actuonix) and USB control via a custom Python wrapper to the C API (available at https://www.actuonix.com/LAC-Board-p/lac.htm). The micromanipulator was controlled using a Python wrapper to the serial control interface. Experiments were monitored live using a custom interface to display trial information, performance, video input, and electrical signals.

A second computer was used to control the two-photon microscope (Sutter Instrument Company, movable objective microscope with resonant scanning) using the MATLAB software ScanImage 2015 (Pologruto et al., 2003) (Vidrio Technologies, RRID: SCR_014307). Excitation light was provided by a Mai Tai Sapphire laser (Spectra-Physics) at 920 nm. A 16x objective lens (Thorlabs, 16X Nikon CFI LWD Plan fluorite objective, 0.80 NA, 3.0 mm WD) was used with ultrasound gel (Sonigel, Mettler Electronics) as the immersion medium. Excitation power measured at the output of the objective lens ranged from 10 to 50 mW. Images were acquired at 28 Hz or 56 Hz, 512 × 512 or 256 × 512 pixel resolution.

Two forms of synchronization signal were sent from the behavior computer to the imaging computer during imaging. The first was a TCP/IP signal indicating animal and session identity. The second was an $I^2C$-based signal routed through a National Instruments card (NI USB-8451). Signals were sent at multiple timepoints throughout each trial, delivering information corresponding to individual defined moments in the trial, which were then embedded in microscope image frames via ScanImage $I^2C$ functionality.

Extracellular recordings were performed using 64-channel Neuronexus silicon probes, which were connected to two amplifier boards (RHD2132, Intan Technologies) using a dual headstage adapter (RHD2000, Intan Technologies). Recordings were made using an Open Ephys acquisition board at a sampling rate of 30 kHz. Similar to imaging, synchronization pulses containing information about timing of the puffs, licks, and rewards were routed through a National Instruments card (NI USB-8451) and connected to the Open Ephys acquisition board using an I/O board (Open Ephys).

## Behavior
### Task

The evidence accumulation task was based on a task used in rats (Brunton et al., 2013). Mice were trained and imaged in 40- to 70 min behavioral sessions, corresponding to 200–300 trials. A session consisted of trials each ranging from 10 to 20 s in duration. In the first phase of the trial, a start tone was presented, followed by a 1 s delay. Next, in the 'cue period,' puffs of air (40 ms duration) were delivered to the whiskers over a period ranging from 1 to 5 s (for all but two sessions, this duration was 1.5 or 3.8 s with 0.15 and 0.85 probability respectively), randomly selected in each trial. The total number of puffs presented was determined by a Poisson process with a rate of 2.5 Hz, a ratio of 1:4 puffs on the two sides, and constrained to a minimum inter-puff interval of 200 ms on each side. In addition to these puffs, bilateral puffs were simultaneously delivered at the start and end of the cue period of every trial. The correct side for a given trial was defined as the side with more puffs, regardless of the generative rates for the two sides. This was followed by a delay period of 200 ms. Then, in the decision phase, the lick ports were brought into the range of the mouse's mouth, and the mouse made a decision by licking the left or right port at any point over a period of 3 s. The decision was defined as the side of the port licked first, regardless of subsequent licks. If the animal licked the correct port, a water reward (4 µL) was immediately dispensed from that port. If the animal licked the incorrect port, an error sound played and no water was delivered. Following the decision was a 3 s phase for the consumption of the reward, if the animal received one. In most experiments, ports were maintained in the lick position for the same duration even in the absence of a reward, but in some sessions, ports were immediately retracted after errors. Following the reward phase was a intertrial interval of 3.5 or 6 s for correct and error trials, respectively. If the mouse made contact with either lick port at any point before the decision phase of the trial, an auditory tone was played, and the trial was cancelled and excluded from analyses. If the mouse made no decision lick, the trial was excluded from analyses.

## Training

Mice were trained to perform the task through a behavioral shaping procedure which typically spanned a period of 10–20 days. In the first stage of training, trials consisted of a 1 s cue period with the same puff rates as the final task, except all puffs following an initial bilateral puff were presented on the correct side only. A water reward was delivered on the correct side in each trial regardless of the actions of the mouse. When a mouse consumed 14 consecutive rewards, it was advanced to the next stage. In the next stage, trial structure remained similar to the previous stage, except rewards were delivered only if the mouse licked the port on the correct side for a given trial at any point in the 3 s response window. In addition, the delay period was extended to 1.3 s during which periodic (2.5 Hz) guide puffs were delivered on the correct side from the end of the cue period until the animal made a decision. To advance, mice completed at least 100 trials at this stage, with 55% correct in a window of 40 consecutive trials, where a correct trial was defined as a match between the side licked first and the correct side for that trial. In the next stage, trial structure remained similar, except bilateral puffs were included at the end of the cue period. In this stage, rewards were only delivered if the first detected lick was on the correct side. As in the final version, error trials included a buzzer sound and a prolonged inter-trial interval. To advance, mice completed at least 200 trials at this stage, with at least 80% correct trials in a consecutive window of 50 trials. In the next stage, the cue period duration was selected randomly as either 2, 2.8, or 3.8 s, with 0.4, 0.3, and 0.3 probability respectively, with a delay period of 200 ms, and guide puffs were no longer delivered. After 100 trials at this stage, and at least 75% correct in a window of 40 trials, mice were advanced. In the next stage, the cue period was its final 3.8 s duration. After a minimum of 125 trials at this stage, and at least 80% correct trials in a window of 25 consecutive trials, mice were advanced to the next stage. In this stage, mice were required to accumulate evidence, with a generative evidence rate ratio of 1:9 for the two sides. When mice completed at least 300 trials and at least 80% correct in 40 consecutive trials, they were advanced to the final task (described above).

Four anti-biasing procedures were implemented during the behavior. First, the probability of drawing a trial on a given side was weighted to produce more trials on the side with worse performance. This mechanism was implemented if in a window of 6 consecutive trials on each side, performance (fraction correct) on one side was 1.5x greater than on the other. In that case, the draw probability of a trial on the worse side was set equal to the ratio of performances on the better to the worse side within the six-trial anti-biasing window. Second, when the same bias metric exceeded 2x, a larger water reward volume (either 1.2 or 1.4 μl, chosen randomly) was given with 60% probability. Third, if bias persisted at the 2x threshold for five consecutive trials, the experimental apparatus was shifted 50 μm toward the side of better performance to bring the stimuli and lick ports of the worse side closer to the animal. At the start of each session, a set of 15–30 warm-up trials were delivered drawing from one or two stages preceding the current one in the shaping procedure, and in warm-up trials of the first four shaping stages, the correct side, instead of being randomized, was alternated. Mice were trained for a maximum of 70 min and a minimum of 40 min unless the animal stopped licking for extended periods of time sooner, in which case they were dismounted early. Occasionally, manual rewards were delivered to encourage licking, and stage advancements or decrements were performed manually to suit animal performance.

## Muscimol inactivation experiments

Mice were trained until they achieved at least 70% correct on the final version of the task in two consecutive sessions. Mice were then subjected to injection sessions, corresponding to saline or muscimol in a randomized order across mice. On injection days, mice were anaesthetized using isoflurane (2–3% for induction and maintenance) for approximately 20 min, during which 100–120 nL of either muscimol (Sigma M1523, 2 mg/ml in sterile saline) or saline were delivered through the injection cannulas (Plastics One, Internal 33GA, 0.9 mm proj) at a rate of 50 nL/min, using an automated injection system (World Precision Instruments Micro4). The injection cannula was left in place for 3 min following each injection to allow diffusion of the solution into the brain. Approximately 1 hr post-injection, mice were mounted on the rig and performed a behavioral session. On at least 2 days prior to the first injection, mice were anesthetized with the same protocol prior to their behavioral sessions in order to acclimatize them to anesthesia. When mice performed below 65% on a session, one or two recovery sessions with no injections were subsequently given such that the animal returned to at

least 70% performance, before any additional injection sessions. Confidence intervals for performance in each session were computed as binomial proportion confidence intervals using the Jeffreys method.

Following the completion of the study (by approximately 3 weeks), mice were injected under the identical protocol with a fluorescent solution (1.8 mg/mL fluorescein (Fisher Scientific S25328) and 0.5 mg/mL CTB-555 (Invitrogen C22843) in sterile saline) to recover the location and spread of the muscimol injection site. After approximately 1 hr, mice were perfused and brains were extracted. Brains were cleared using iDISCO (*Renier et al., 2014*) and imaged on a lightsheet microscope (LaVision BioTec Ultramicroscope II, 488 nm excitation laser, 514/30 nm emission filter (Semrock FF01-514/30-25)).

## Data analysis

### Software
Data analysis was performed using custom analysis packages written in Python (RRID:SCR_008394) 3.5 and 3.6 (https://github.com/bensondaled/pyfluo; *Deverett, 2018b*; copy archived at https://github.com/elifesciences-publications/pyfluo), which make use of Numpy 1.12.1 and Scipy 0.19.1 (*van der Walt et al., 2011*), Pandas 0.20.3 (*McKinney, 2010*), Matplotlib 2.0.2 (*Hunter, 2007*), IPython 6.1.0 (*Perez and Granger, 2007*), Scikit-learn 0.18.1 (*Pedregosa et al., 2011*), Scikit-image 0.13.0 (*scikit-image contributors et al., 2014*), OpenCV 3.3.0 (*OpenCV team, 2017*), Statsmodels 0.8.0 (*Perktold et al., 2017*), and OASIS (*Friedrich et al., 2017*).

### Psychometrics
*Figure 1B* shows psychometric performance data for individual mice and for the 'meta-mouse,' consisting of pooled trials from all trained mice. Data for psychometrics were obtained only from trials in the accumulation stages of the task, and not from the preceding stages during the shaping procedure. All analyses contain only trials in which mice made decision licks, such that incorrect trials correspond to licks in the wrong direction, and never the absence of licks. Psychometric curves in *Figure 1—figure supplement 2D* were fit to a four-parameter logistic function of the form:

$$y(x) = y_0 + \frac{A}{1 + e^{\frac{-(x - x_0)}{b}}}$$

### Behavior regression analysis
To determine the dependence of animal choice on stimuli in different temporal bins of the cue period, we performed a regression-based analysis. A logistic regression was computed, with animal decision on a trial-by-trial basis as the predicted variable. The input for each trial was a vector of 5 values, corresponding to the difference in right vs left puffs in five temporally uniform bins of the cue period. Data for regression analysis consisted of trials with the primary cue period duration of 3.8 s. In the 'shuffle bins' control, the vector of R-L values was shuffled across bins for each trial. In the 'shuffle choices' control, the choices of the animal were shuffled across trials.

### Behavioral regression model
Effects of the muscimol inactivation were assessed by fitting behavioral data to a regression model (*Busse et al., 2011*; *Licata et al., 2017*) that considers four factors which contribute to the animals' decisions: stimulus strength, $s$; bias; success (correct decision) on the previous trial, $h_{success}$; and failure (incorrect decision) on the previous trial, $h_{failure}$. Stimulus strength $s$ was defined as the #R-#L puffs scaled from -1 to 1. The history term $h_{success}$ was 0 if the previous trial was incorrect and -1 or 1 if the previous trial was a correct left or right choice respectively. $h_{failure}$ was 0 if the previous trial was correct and -1 or 1 if the previous trial was an erroneous left or right choice respectively. The decision was modeled as a random variable such that the probability $p$ of a rightward choice in the current trial was modeled as

$$ln\left(\frac{p}{1-p}\right) = b_{evidence\ sensitivity}s + b_{success\ history}h_{success} + b_{failure\ history}h_{failure} + b_{bias}$$

The four coefficients $b_{evidence\ sensitivity}$, $b_{success\ history}$, $b_{failure\ history}$, $b_{bias}$ were fit separately for each

muscimol session and control session directly preceding it, by a binomial generalized linear model with a logit link function. To compute the significance of the behavioral effects caused by muscimol inactivation, each of the four best-fit parameters from the muscimol trials was compared to the best fits for that parameter in 10,000 bootstrapped datasets from the baseline trials, and the p-value was computed as the fraction of instances in which the muscimol fit exceeded the baseline fit.

## Imaging data preprocessing

Imaging data were first motion corrected using a custom template matching-based procedure, then regions of interest (ROI) were selected manually based upon cell morphology and calcium activity. The manually selected dendritic ROI were subsequently refined using the following procedure: for each ROI, the mean activity of all pixels inside the ROI was extracted for each frame in the behavioral session, yielding a single time series. The Pearson correlation coefficient was computed between this time series and every pixel in the dataset. These correlation coefficients were assembled into a correlation coefficient image with the same dimensions as an imaged frame, and regions of interest were automatically extracted from this image by applying a median filter, followed by thresholding the image at two standard deviations above the mean, then detecting and selecting the connected component that most overlapped with the manual ROI. The resulting ROI was excluded if it contained less than 0.5 or greater than 3.5 times the number of pixels in the manually selected ROI. After all ROI were refined in this way, a second processing step merged overlapping and highly correlated ROI to ensure that the resulting ROI corresponded to unique physiological sources. First, mean activity time series traces were extracted for each refined ROI. Next, the pairwise correlation coefficients of traces from each ROI were computed, and an undirected graph was constructed where each ROI composed a node, and edges were placed between nodes with correlation values exceeding a threshold of 0.5. Next, connected component clusters in the graphs with greater than one node were subjected to the following procedure: for each pair of ROI within the cluster, the two ROI were merged if >50% of either ROI's pixels was contained within the other's boundaries. A similar graph-based approach was used to detect neighboring ROI with high correlations corresponding to a single cellular origin; if the correlation values of two ROI exceeded 0.8 and they fell within 5 micrometers of each other, they were merged. Finally, pixels within the motion boundaries of the motion-corrected images were removed from all ROI.

For each ROI, the mean activity of all pixels in the ROI was computed for each frame, yielding raw time series data. $\Delta F/F_0$ was then computed from these raw traces, with baseline $F_0$ being computed as the minimum of a median-filtered (1 s kernel) 12 s sliding window preceding each time point. Somatic data analysis was restricted to trials with matching 3.8 s cue periods, which comprised 85% of trials. Dendritic data analysis included trials of all cue period durations.

## Modulation index r

Modulation index r was defined as Pearson's correlation between the somatic calcium signal of a given cell and a uniform time vector ('cue-period time'). For trial-by-trial analyses (choice and evidence decoding, linear model to evaluate puff-fluorescence relationship), $r$ was computed on a trial-by-trial basis. For other analyses, $r$ was computed over the mean somatic signal in the specified time period. For comparison of $r$ in the pre-cue period and cue period, $r$ was computed over a 2 s period immediately preceding, or in the middle of the cue period, respectively. For all other analyses, $r$ was computed over the duration of the cue period. Significance of this modulation was computed by comparing the fraction of cells for which $|r_{cue\ period}|$ exceeded $|r_{pre-cue\ period}|$. The 95% confidence interval for this fraction was computed using bootstrapping: cells were randomly sampled with replacement to create resampled datasets (n = 10,000), and the same fraction was computed for each of these datasets. Boundaries of the confidence interval were then computed as the 5th and 95th percentile of these bootstrapped fractions. Return to baseline was computed by comparing the mean activity of each cell before the cue period and after the inter-trial interval; in total, 198,542 such comparisons were made for all cells in the study.

## Analysis of electrophysiological recordings.

High-pass filtering of the raw data at 300 Hz, common median referencing, and automatic spike sorting was achieved using Kilosort (*Pachitariu et al., 2016*; *cortex-lab, 2018*; https://github.com/

cortex-lab/Kilosort). Spikes were further sorted manually using Phy (https://github.com/kwikteam/phy; **kwikteam, 2018**). Instantaneous firing frequency was used to determine the firing rate. Purkinje cells were identified based on their firing rate and the occurrence of complex spikes on either the same channel or on neighboring channels.

### Puff-triggered responses
Puff-triggered responses were computed by averaging the ΔF/F signal of a given region of interest triggered to the onset of individual puffs, drawn from trials with sparse (<3) puffs on the given side, and excluding bilateral start and end puffs. Mean $t_{t/2 \ decay}$ of the puff-triggered response was computed as the time elapsed from the peak value of the mean response to L and R puffs until it fell 50% of the way to baseline, where baseline was defined as the mean value of the response in the 400 ms preceding the puff.

### Choice and evidence decoding
Somatic activity or behavioral movie measurements were used to decode the choice or correct side (evidence) on a trial-by-trial basis. Each trial was segmented into 400 ms time bins, in which the mean activity of each cell (somatic decoding) or the mean movement index for each movement feature (behavioral decoding) was computed. For each time point, the vector of these mean values was used to predict the behavioral variable (upcoming choice or side with more total evidence at end of trial) by logistic regression. Each regression was scored by k-fold cross validation with k = 5. Because choice and evidence are correlated in the behavioral task, we employed a procedure to investigate their effects separately. First, to determine whether evidence-related information was contained in the neural activity independently of choice-related information: we compared evidence-decoding accuracy with accuracy on a shuffled dataset in which the side of evidence on each trial was randomly assigned to another trial in which the same choice was made as the original trial. In this shuffled dataset, all evidence-decoding accuracy arises from correlated choice-related information, since evidence-related information was removed; therefore, any excess decoding performance relative to this baseline exhibited by the unshuffled dataset arises from independent evidence information. We also did the converse to test for independent choice-related information: that is, shuffled choice across trials within the same evidence category (where evidence was binned into four groups based on the magnitude [#R-#L] of evidence, to account for the relationship between evidence magnitude and choice probability), and evaluated choice decoding accuracy. These 'independent' shuffles constitute the 'Ind' condition for choice and evidence decoding. In addition, complete shuffles were performed in which choice/evidence was shuffled across trials with no regard for the other variable. These shuffles constitute the 'Shuffle' condition.

### Linear model for evaluating puff-fluorescence relationship
To determine the relationship between somatic signals and puff number on a trial-by-trial basis, a two-factor linear model was constructed for each cell. The factors corresponded to the number of left-sided and right-sided puffs in each trial, and the dependent variable to the mean pre-decision fluorescence (500 ms preceding the decision phase) of the given cell in each trial. The best-fit coefficients were interpreted as the relationship between fluorescence and puff number for that cell. Significance of these coefficients was scored by the p-value associated with each coefficient in the model (two-tailed unpaired t-test of regression weight values). Cells with p<0.05 for a given coefficient were considered to have ramps significantly correlated with puffs on the corresponding side. Significance cells were grouped into categories according to the sign of the coefficient (+ or –), and the side of the puffs with significant coefficients (L, R, or LR for both). In the shuffle condition, the puff numbers were shuffled across trials while holding the animal's choice constant, and the same analysis was run.

### Movie-based behavior measurements
Movements of the mouth, whiskers, nose, and paws were made using the behavioral movies acquired during all behavioral sessions in which calcium imaging was performed. For each movie from each animal, regions of interest (ROI) were manually selected corresponding to the aforementioned regions in the field of view. Full-session traces were extracted as the mean pixel value within

the ROI in each frame. Mouth movements were measured using the laterally positioned camera, where deviation of the mouth and chin was best detected. Full-session traces corresponding to mouth movement were mean-subtracted and normalized from 0 to 1. For those measurements, the normalized licking measure was computed as the absolute value of the mouth-movement trace, with a median-filtered (0.5 s kernel) drifting baseline signal subtracted off, normalized from 0 to 1. Deviations in overall image brightness were normalized by a fiducial measurement in the field of view not involving the animal that reliably tracked field-of-view brightness. The remaining movement measurements (nose, left whiskers, right whiskers, left paw, and right paw) were extracted from the camera positioned below the animals' faces. For these measurements, traces were extracted in the same manner as above, and the normalized movement index was computed as absolute value of the trace derivative, normalized from 0 to 1.

### Dendritic events

Dendritic events were detected using the OASIS autoregressive deconvolution algorithm (*Friedrich et al., 2017*), with AR(1) and an $L_0$ sparsity penalty, and binarized with a threshold of 0.1.

### Analysis of dendritic responses

To test the hypothesis that dendritic signalling encoded motor events, we computed the mean dendritic response to a number of specific motor actions, and compared this response across error and correct contexts on a cell-by-cell basis. In these analyses, lick initiation ('start licking') and cessation ('stop licking') were defined as licks that were preceded or followed by, respectively, at least 250 milliseconds without licking on that side. For the analysis in *Figure 4F*, activity was measured in a 1-second window centered at this event. In addition, supplementary analyses measure the signal instead in the 200 ms preceding or following the event to account for the possibility that signals encode specifically preparatory or response signals to the events, respectively (*Figure 4—figure supplement 1*). For licking magnitude analyses, trials were categorized according to the quantity of licks in the same time window in which the dendritic response was measured (*Figure 4E*), and the dendritic response was measured in the same manner as the data in *Figure 4E*. For analyses of orofacial movements, the onset times of distinct movements in each orofacial region were determined, according to frames at which the time series data for that feature exceeded 2.5 standard deviations above the mean, and activity was measured in the one-second window centered at these events.

For all of the above movement measurements, movement events were split into those which occurred in error contexts (i.e. the time window for which the dendritic response was measured for *Figure 4E*, in error trials), and those which occurred outside error contexts (in correct trials). Mean dendritic activity events were computed for both contexts for every movement measurement, and the error:correct activity ratio was then computed for each individual cell. To compare dendritic responses at varying trial difficulties, the difference was computed on a session-by-session basis between the mean dendritic response in the strongest and weakest evidence (|#R-#L|) trials, and the significance of this difference was compared using a two-tailed paired t-test.

### Outcome decoding

Dendritic activity was used to decode the outcome (correct or error) on a trial-by-trial basis. For each cell in each trial, the post-decision response was computed as the mean $\Delta F/F$ value in the 800 ms following ('Post-choice') or preceding ('Pre-choice') the decision. The outcome of each trial was predicted by logistic regression using these values as predictors. For each session analyzed, 1000 batches of trials were randomly sampled in which the number of correct and error trials were matched, and a regression was performed on each, scored by k-fold cross validation with k = 3. In the shuffle conditions, the outcomes of the trials were shuffled and the decoding accuracy on the shuffled dataset was reported.

## Acknowledgements

We thank the members of the laboratories of SW, David Tank, Ilana Witten, and Carlos Brody for discussion and technical assistance, as well as David Tank for advice and support, Esteban Engel for viruses, Ben Scott, Ilana Witten, Carlos Brody, Sandra Aamodt, and Alex Riordan for comments on

the manuscript, Stephan Thiberge for microscopy, Steve Lowe for machine shop assistance, Jess Verpeut and Tom Pisano for technical help, and Sarah Welsh for data analysis. Funded by National Institutes of Health grants R01 NS045193, U01 NS090541, U19 NS104648, R01 MH115750, and F30 MH115577, and the Nancy Lurie Marks Family Foundation.

## Additional information

### Funding

| Funder | Grant reference number | Author |
|---|---|---|
| National Institute of Mental Health | MH115577 | Ben Deverett |
| National Institute of Neurological Disorders and Stroke | NS090541 | Ben Deverett<br>Sue Ann Koay<br>Samuel S-H Wang |
| National Institute of Neurological Disorders and Stroke | NS104648 | Ben Deverett<br>Sue Ann Koay<br>Samuel S-H Wang<br>Marlies Oostland |
| National Institute of Mental Health | MH115750 | Samuel S-H Wang<br>Marlies Oostland |
| National Institute of Neurological Disorders and Stroke | NS045193 | Samuel S-H Wang |
| Nancy Lurie Marks Family Foundation | | Samuel S-H Wang |

The funders had no role in study design, data collection and interpretation, or the decision to submit the work for publication.

### Author contributions

Ben Deverett, Conceptualization, Resources, Data curation, Software, Formal analysis, Investigation, Visualization, Methodology, Writing—original draft, Writing—review and editing; Sue Ann Koay, Formal analysis, Methodology, Writing—review and editing; Marlies Oostland, Formal analysis, Investigation, Writing—review and editing; Samuel S-H Wang, Conceptualization, Resources, Supervision, Funding acquisition, Writing—review and editing

### Author ORCIDs

Ben Deverett (iD) http://orcid.org/0000-0002-3119-7649
Marlies Oostland (iD) http://orcid.org/0000-0001-9474-4040
Samuel S-H Wang (iD) http://orcid.org/0000-0002-0490-9786

### Ethics

Animal experimentation: Experimental procedures were approved by the Princeton University Institutional Animal Care and Use Committee (protocol #1943-16) and performed in accordance with the animal welfare guidelines of the National Institutes of Health. All surgery was performed under isoflurane anesthesia and suffering was minimized in all ways possible.

### Decision letter and Author response

Decision letter https://doi.org/10.7554/eLife.36781.021
Author response https://doi.org/10.7554/eLife.36781.022

## Additional files

### Supplementary files

• Transparent reporting form

DOI: https://doi.org/10.7554/eLife.36781.018

**Data availability**

The data for the main figures are available via the GitHub repository https://github.com/wanglab-princeton/accumulating_puffs (copy archived at https://github.com/elifesciences-publications/accumulating_puffs). The complete raw data are available from the authors upon request.

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
