## [Decision Letter]

Thank you for submitting your article "A cerebellar role in evidence-guided decision-making" for consideration by *eLife*. Your article has been reviewed Richard Ivry as the Senior Editor, a Reviewing Editor, and three reviewers. The reviewers have opted to remain anonymous.

The reviewers have discussed the reviews with one another and the Reviewing Editor has drafted this decision to help you prepare a revised submission.

Summary:

This is an interesting study describing a potential role for the lateral posterior cerebellum in a novel decision-making task. The cerebellum has received much less focus than cortex in decision-making research despite clear evidence from, among other sources, human imaging and lesion studies that the cerebellum is involved in a wide range of cognitive processes. There is also a rich tradition of theoretical modeling inspired by the cerebellum's unique circuit organization and role in sensorimotor coordination that appears potentially relevant to many cognitive operations, raising interesting questions about what the cerebellum might contribute to decision-making. Therefore, the approach and data reported in the current manuscript are promising and could provide meaningful novel insight into the neural basis of decision-making. Despite this potential, and that it does report some interesting observations, the main limitation of this work is that it is hard to say what the cerebellum is doing during the task. In particular, it is not clear that it is possible to conclude that the cerebellum is truly involved in evidence accumulation. The concern about the support for evidence accumulation holds for the inactivation experiments as well as for the imaging.

Essential revisions:

1) Inactivation results: Inactivation results are suggestive of an important role for the cerebellum in this task. However, from the data presented, it is not clear that it is possible to conclude that the cerebellar inactivations are interfering with evidence accumulation itself, and not just somatosensation or motor output. At a minimum, the authors should show psychometric curves for the 5 individual muscimol-treated animals. Assessing the effects on the slopes and offsets of the individual psychometric curves might be more revealing about which aspects of task performance were affected by inactivation of crus I. Further, there is the possibility that with this experimental design, muscimol inactivation may not be adequate to determine whether the cerebellum is accumulating evidence, because changes in the psychometric function can also be explained by a decrease in sensory sensitivity alone. Careful modeling of the data may help.

2) Evidence for evidence accumulation signals from imaging of somatic calcium in Purkinje cells: The authors report that appear similar to evidence-dependent and choice-selective ramping responses previously reported in many cortical and subcortical areas of primates and rodents. The manuscript reports temporal dynamics of Purkinje cell somatic calcium signals that, in many cells, manifest as a gradual ramping response during the cue period that is correlated with both the number of pulses on one or the other side (evidence) and/or the animal's eventual response (choice). These responses are interpreted with analogy to the ramping responses in trial-averaged spike rates recorded from parietal cortex (along with other forebrain regions) that are thought to encode the time-integral of sensory evidence. However, the manuscript does not convincingly demonstrate that cerebellar neurons are actually encoding accumulated evidence. In general, given the uncertainty about the origin of these signals, more caution is warranted in interpretation of the apparent ramping signals. Specifically:

a) While there appear to be lots of time-modulated signals, the link to evidence accumulation and choice is not clear. It is apparent that the most salient feature of the recorded cerebellar population is a constant modulation, either upwards or downwards, with the passage of time in the trial. The presence of this signal complicates the interpretation of any ramping activity as being attributable to evidence accumulation. However, the statistical analyses (i.e. the linear model reported in subsection “Neuronal signatures of choice and evidence in Purkinje cells”) ignore this component of the responses. This might be valid if the representation of time were truly independent from the representation of evidence. But no such independence is established. For example, cells that are more strongly driven by evidence might also be more strongly driven by time, or more likely to be driven by time in one direction or another. The authors should more carefully consider the relationship between the different components of the cerebellar responses and/or formally control for potentially confounding effects of the time-related responses in the statistical models.

b) Given the temporal resolution of the calcium indicator, it is difficult to interpret any evidence-related signal dynamics as reflecting an underlying ramp in neuronal firing rate. The original description of the use of somatic calcium imaging to track changes in Purkinje cell simple spike rate (Ramirez and Stell, 2016) found that these signals were so slow that step changes in simple spike rate could result in ramps of fluorescence like those seen here. This technical concern would best be addressed by at least some recordings comparing the calcium signals to simple spike rate with electrophysiology. Although the authors have tried to be careful in their writing, the text currently leaves plenty of room for misinterpretation by less technical readers. They should clarify the text further to be much more clear about the limitations of interpreting the temporal dynamics of the calcium signals.

c) The relevant decision variable in the pulse accumulation task is the *difference* in pulses between the two sides. The authors seem to know this point well based on how they plot the psychometric functions. Yet at best a small minority of Purkinje cells encode this value. Subsection “Neuronal signatures of choice and evidence in Purkinje cells” reports that 39/843 cells (less than 5%) encode either a sum or difference of the number of pulses on each side. The specific figure for the number of cells encoding a difference is not reported, but Figure 3 suggests that it is about 5 cells (so ~2% of evidence-modulated cells or ~0.5% of all cells). These numbers question strong conclusions about the representation of the decision variable and accumulated evidence in cerebellum. Further, this result could be taken to show that responses in rodent cerebellum are different from those in primate cortex that are often interpreted as a representation of the decision variable for perceptual discrimination. Notably, it also indicates a dissociation between rodent cerebellum and neocortex, as Scott et al., estimate that as many as 1/3 of evidence-modulated cortical cells are better explained by the difference between sides. Therefore, the data strongly suggest that the critical decision-making operations are actually implemented downstream of the recorded cerebellar population. This strikes me as highly relevant to how the overall results should be interpreted, and it should be emphasized more strongly in the summary and discussion.

d) The analysis of the representation of accumulated evidence (subsection “Neuronal signatures of choice and evidence in Purkinje cells” and Figure 3) ignores the time course of representation, focusing on a short window immediately before the decision. Even if we disregard technical problems explained above in 2a-c, it is unclear whether the cerebellar neurons represent integration of evidence over time or merely its final outcome. Note that the population analysis for the representation of evidence (Figure 2F) does not answer this question (and is generally not quite informative) because evidence is defined as the "correct" choice in that analysis rather than the magnitude of evidence (#R-#L).

3) "Error-related" dendritic responses

In addition to somatic calcium signals, the authors analyzed dendritic calcium signals and found that these were higher on error vs. correct trials. The implications of this observation are emphasized heavily; for example, the Discussion section concludes that "the cerebellum, which learns from error to guide action, may help in the learning and tuning of accurate responses". This is an interesting proposal, but one might have had the same belief prior to seeing the results reported here (given existing human data and theories about the cerebellum), and it's not clear how the data should update it.

a) There is speculation about how the observed error responses could be used as error signals to guide learning, in line with existing models of cerebellar learning. However, it was not clear to the reviewers how that would work, especially given that the error responses observed were not directional. Typically, Purkinje cell complex spikes are thought to provide a directional signal for learning, not just a correct/ incorrect signal. There are no analyses to support the proposed functional role of these error related responses as being involved in "tuning" responses or correcting errors. Yet, it should be possible to provide a more thorough analysis of the error signals:

What do error signals predict about future behavior? An obvious analysis would be to ask whether the magnitude of the error signal on trial n−1 influences choice accuracy on trial n. This seems to be a clear prediction from the proposal that the cerebellum "tunes" behavior or "corrects errors".

c) How do the error signals relate to the strength of evidence on each trial? This is briefly mentioned at the end of the Results section and in Figure 4—figure supplement 1, but without any statistical tests or interpretation. Whether and how the error signals are modulated by the strength of evidence seems key to determining their functional role in updating behavior, if any. In particular, we would have expected that true "error" responses would be higher for easier trials, but the opposite appears to be true.

4) Ruling out sensory and motor confounds as potential sources of somatic and dendritic calcium signals

a) Figure 3B indicates that isolated puffs produce a calcium response that gradually rises and falls over ~1 s. On trials with strong evidence, these signals will overlap. Assuming that they sum reasonably linearly, overlapping calcium responses from neurons that encode only the transient presentation of evidence would nevertheless give the impression of a gradual ramp with a slope that depends on the quantity of evidence. Note that this is a different issue from the point raised in the Discussion section about distinguishing single-trial ramps from steps. Rather, it makes it unclear whether the evidence-related cerebellar responses correspond to a representation of the momentary sensory evidence or to the magnitude of accumulated evidence that drives choice. This distinction is critical to interpreting proposed neural implementations of evidence accumulation models. One possible way to address this would be to record responses during stimulus presentation from animals that are not engaged in making a decision.

b) The authors attempt to rule out that the somatic signals are related directly to movement of the animal and less related to the "cognitive" variables. The analysis in the supporting figures is in line with the level of detail often applied in the field currently. But, is it possible that the activity is related to other movements of the animal rather than in the orofacial region? The authors should either cite strong evidence showing that these are the only relevant regions for the area of cerebellum examined or provide additional videography results from other parts of the body (e.g. paws). This is of course a significant point to make as strong as possible given the cerebellum's long-established role in motor behaviors.

c) The analyses in Figure 4—figure supplement 1 do not convincingly rule out the possibility that the difference in dendritic calcium signals on error trials resulted from differences in licking on error trials. That figure clearly shows that cessation of licking proceeds at different rates on error trials compared to other times. How does trial type (correct vs error) affect the relationship between licking (or changes of rate in licking) and dendritic calcium signals?

[Editors' note: further revisions were requested prior to acceptance, as described below.]

Thank you for resubmitting your work entitled "A cerebellar role in evidence-guided decision-making" for further consideration at *eLife*. Your revised article has been favorably evaluated by Richard Ivry (Senior Editor), Megan Carey (Guest Reviewing Editor), and two reviewers.

The manuscript has been improved but there are some remaining issues that need to be addressed before acceptance, as outlined below:

The authors have made a concerted effort to clarify the text of their paper and have provided reasonable responses to many of the points raised. As was noted in the first round of reviews, rebuttal, and in the text, it remains unclear what the cerebellum's specific role or computation is during the task, and the reviewers were somewhat disappointed that most of the key questions are deferred to future studies. However, the reviewers also felt that defining cerebellum's exact role in the task may be asking too much for a first study, particularly given the limitations of the dataset and the conceptual and analytical complexities that prevent the authors from specifying the nature of neural representations and the source of the behavioral deficit following cerebellar inactivation.

Overall, there are several interesting leads about the role(s) of cerebellum in perceptual decisions in this paper. While none of these are firmly established by the current study, the reviewers agree that the writing and presentation of the results is generally fair and not greatly over-stated, and favor publication so that the results can be evaluated by the field and followed up in future studies from other groups.

The reviewers note that in the process of clarifying what can and cannot be claimed based on the existing data, the scope of the paper is limited to three points: successful training of mice to do the task, reduced accuracy following cerebellar inactivation, and representation of task-relevant variables in cerebellar neural population without specifying the exact nature of the represented variables. Before publication, the reviewers agree that it is important to ensure that after toning down its claims, the manuscript does not leave behind any statements that could mislead readers as to what is actually shown. The reviewers have identified the following statements from the Title, Abstract, and Discussion section that are potentially misleading and should be restated with more specific statements that more accurately match the conclusions that can be supported by the data.

Title

A cerebellar role in evidence-guided decision-making.

Impact Statement

The lateral posterior cerebellum participates in evidence-accumulation-based decision-making, and Purkinje neurons in this region encode choice-, evidence-, and error-related variables. [Suggest replacing this stronger statement with language more like that used in the rebuttal, such as "choice- and evidence-related information is present in lateral posterior cerebellum and could participate in decision-making computations during a decision-making task involving evidence accumulation."].

Abstract

- Here we show that during perceptual decision-making over a period of seconds, decision-, sensory-, and error-related information converge on the lateral posterior cerebellum in crus I, [The presence of task-related signals is shown. Convergence is not, and decision and sensory signals are not clearly dissociated].

- Demonstrated that cerebellar inactivation reduces behavioral accuracy without impairing motor parameters of action [Not all motor parameters were controlled for].

- We found that Purkinje cell somatic activity encoded choice- and evidence-related variables [Please avoid the suggestion that the specific variables that are encoded have been determined].

- Decision errors were represented by dendritic calcium spikes, which are known to drive plasticity [This could misleadingly suggest that they are known to drive plasticity in this context].

- We propose that cerebellar circuitry may contribute to the set of distributed computations in the brain that support accurate perceptual decision-making. [Should be more focused on task performance].

Discussion section

- Cerebellar inactivation reduces animals' use of evidence and increases their use of choice history. [given the limitations of the interpretation of the inactivation experiments, this statement should be more conservative].

- Given the temporal resolution of calcium measurements, our somatic signals may correspond to firing rate ramps (Shadlen and Newsome, 2001), steps (Latimer et al., 2015), or more complex response profiles that form a temporal basis for evidence accumulation (Scott et al., 2017). [This statement, as well as the corresponding section of the Results section, should include an explicit reference to the time course of somatic calcium signals from Purkinje cells, which is at least an order of magnitude slower than typical calcium imaging (Ramirez and Stell, 2016).].

-The task-modulated activity we observe encodes both choice-related and evidence-related variables that may be used during the decision-making process. [Please avoid the suggestion that the specific variables that are encoded have been determined].

- We observed an excess of dendritic calcium events coincident with decision errors, demonstrating for the first time observations compatible with error-associated signalling in a decision-making reward context. [Given the limitations of the interpretation of these signals, this statement should be more conservative].

[Editors' note: further revisions were requested prior to acceptance, as described below.]

Thank you for resubmitting your work entitled "Cerebellar involvement in an evidence-accumulation decision-making task" for further consideration at *eLife*. Your revised article has been favorably evaluated by Richard Ivry (Senior Editor), and Megan Carey (Guest Reviewing Editor).

The manuscript has been improved but there are some final issues that need to be addressed before acceptance, as outlined below:

In response to the request to re-evaluate the evidence for ramping signals that could be obtained with the somatic calcium imaging, the authors now state (Subsection “Purkinje cell somatic calcium encodes task-relevant information”), "Therefore our observed increasing and decreasing time courses of calcium could reflect various firing rate profiles, such as impulse responses, ramps, or steps."

In light of this revision, as well as the fact that the electrophysiological evidence provided in Figure 2—figure supplement 2 is from only a few cells, all of which show positive ramps of activity, the following statements should also be revised:

- (Subsection “Purkinje cell somatic calcium encodes task-relevant information”), "We did find that electrically recorded Purkinje cells exhibited gradually increasing rates of firing throughout the cue period (Figure 2—figure supplement 2), suggesting that on average across trials, the fluorescence signals we observed correspond to firing rate ramps."

- (Discussion section), "Our electrical recordings also showed ramps, suggesting that temporally filtered firing rate ramps are sufficient to account for our observed fluorescence signals."

In both instances, we suggest removing the second clause, starting with "suggesting that…"

[Editors' note: further revisions were requested prior to acceptance, as described below.]

Thank you for resubmitting your work entitled "Cerebellar involvement in an evidence-accumulation decision-making task>" for further consideration at eLife. Your revised article has been favorably evaluated by our editors again, but there remain some issues that need to be addressed before acceptance, as outlined below. Given that this is the third request for revisions, we will be unable to follow with any more. Please attend to this final issue one way or the other so that the next letter will be the final one.

We appreciate the authors' desire to speculate here. However, in our view, the "suggests/ consistent with" was not the only problem with this sentence. There is also a problem with "sufficient". The electrophysiological evidence in Figure 2—figure supplement 2 is anecdotal and non-quantitative. For this statement to be left in, it would need to be adequately supported. In our view, this would require a quantitative comparison between imaging and electrophysiology results. In particular, we would want to know:

- How many Purkinje cells in total were recorded from electrophysiologically? How many of these showed ramping? (all of the cells they showed us show positive ramps, but it is not clear if those were selected from a larger data set)

- Did any Purkinje cells show ramping calcium signals without a transient increase in firing rate?

- Did any Purkinje cells show ramping calcium signals without ramps in firing rate (for instance, in cases where only a transient increase in firing may have been observed electrophysiologically)?

- Why are no decreasing activity ramps found with electrophysiology, but they are found with imaging?

- What accounts for the decreasing ramps that were observed with calcium imaging?

- What would the predicted calcium signals be for the examples shown if the spike rates recorded electrophysiologically (with and without the transient increase/ ramping components) were convolved according to Ramirez and Stell (2016)? And/or with the authors' own convolution/ deconvolution methods, from the simultaneous calcium imaging/ electrophysiological recordings that they performed?

We give the authors the choice of either fully addressing these points, or using compromise language, such as "Preliminary electrical recordings also showed ramps, consistent with the idea that temporally filtered firing rate ramps may account for the observed fluorescence signals."

---

## [Author Response]

Summary:This is an interesting study describing a potential role for the lateral posterior cerebellum in a novel decision-making task. The cerebellum has received much less focus than cortex in decision-making research despite clear evidence from, among other sources, human imaging and lesion studies that the cerebellum is involved in a wide range of cognitive processes. There is also a rich tradition of theoretical modeling inspired by the cerebellum's unique circuit organization and role in sensorimotor coordination that appears potentially relevant to many cognitive operations, raising interesting questions about what the cerebellum might contribute to decision-making. Therefore, the approach and data reported in the current manuscript are promising and could provide meaningful novel insight into the neural basis of decision-making. Despite this potential, and that it does report some interesting observations, the main limitation of this work is that it is hard to say what the cerebellum is doing during the task. In particular, it is not clear that it is possible to conclude that the cerebellum is truly involved in evidence accumulation. The concern about the support for evidence accumulation holds for the inactivation experiments as well as for the imaging.

We thank the reviewers and editor for their thoughtful comments on the manuscript. We have added new data, analyses, and substantial clarifications that we think improve the quality of the study. In some cases, we think the objections raised by reviewers were consequences of our failure to properly convey the scope of our study.

We fully agree that a tremendous amount stands to be resolved with respect to the cerebellar role and computations in our decision-making behavior. Our findings are at an early stage relative to the decision-making field. It is not our intent at this time to dissect precisely the evidence integration process as many studies have, but rather to study the whole perceptual decision-making process with a more agnostic view about cerebellar roles. We cannot yet make strong claims about the particular computational aspect of decision-making to which the cerebellum contributes. In light of this and of the reviewer suggestions, we have emphasized more strongly the extent to which many questions remain unanswered.

Of specific note with respect to a number of reviewer comments: we do not interpret our findings as evidence that the cerebellum performs the integration computation in evidence accumulation. Furthermore, we are aware that the particular roles of brain regions in the evidence-accumulation process are intricate, deeply studied, and yet still unresolved (Brody and Hanks, 2016).

In that light, our goal was to probe for an involvement of cerebellar activity in the perceptual decision-making process, which to the most basic extent has not been previously shown. We asked three fundamental questions: (1) Can we train mice on a new evidence-accumulation decision-making task? (2) Does cerebellar inactivation affect behavioral performance? (3) Are task-relevant variables represented in any way in cerebellar neuronal populations? We reasoned that these three demonstrations would constitute novel and interesting findings and would be a basis for substantially larger-effort endeavors (which we are now pursuing).

To make this clearer in the manuscript, we have adjusted the text to emphasize the result: i.e. demonstration of a cerebellar role in perceptual decision-making, using evidence accumulation as a paradigm, but where the cerebellar role may or may not be directly in the computations of updating an evidence integrator per se. We were careful to present it this way in the manuscript and have attempted to clarify it more. We consider follow-up studies with detailed high-resolution measurements and inactivations an important and interesting next step, and we are pursuing that.

Essential revisions:1) Inactivation results: Inactivation results are suggestive of an important role for the cerebellum in this task. However, from the data presented, it is not clear that it is possible to conclude that the cerebellar inactivations are interfering with evidence accumulation itself, and not just somatosensation or motor output.

We view the inactivation experiment as an initial demonstration of cerebellar necessity, and not a comprehensive modelling effort. Rather, we add a new region of interest to the emerging brainwide view of perceptual decision-making, noting that the inactivations demonstrate a causal role for the cerebellum in effectively converting sensory evidence into a decision at some stage of that complex process. This could correspond to sensory input gain biases, post-categorization biases, or other non-accumulation-specific biases. We have now noted this important distinction more clearly in the Discussion section. Below we address our thoughts on the possibilities of specific roles for the cerebellum but we do not believe to have resolved those yet using this study.

First, to address the possibility of interference with motor output: we were particularly mindful of the possibility of a strictly motor deficit, and we believe the licking measurements and control trials we included (Figure 1—figure supplement 2B) address this.

Regarding the possibility of sensory deficits and other non-accumulation impairments: as we explain in the initial section of our response, we completely agree: the evidence-accumulation process, as opposed to related sensory gain and premotor processes, is difficult to pin down and we are aware of recent findings in rodent decision-making which indicate non-accumulation roles for regions that have been long studied in decision-making (Erlich et al., 2015; Katz et al., 2016). We do not intend to claim that the cerebellum performs specifically evidence integration.

At a minimum, the authors should show psychometric curves for the 5 individual muscimol-treated animals. Assessing the effects on the slopes and offsets of the individual psychometric curves might be more revealing about which aspects of task performance were affected by inactivation of crus I. Further, there is the possibility that with this experimental design, muscimol inactivation may not be adequate to determine whether the cerebellum is accumulating evidence, because changes in the psychometric function can also be explained by a decrease in sensory sensitivity alone. Careful modeling of the data may help.

In the submitted version of the manuscript, the evidence sensitivities, biases, and trial history effects of individual animals were represented by the session-by-session fits in the logistic regression model shown in Figure 1—figure supplement 2C (and subsection “A decision-making task for cerebellar investigations”). This behavioral model has been used in multiple published decision-making studies, it considers more variables than a psychometric fit does, and it provides a highly useful view of the effects introduced by the inactivation.

Psychometric curves

Nevertheless, we now include psychometric curves that similarly demonstrate the reduced sensitivity presented quantitatively in the aforementioned modelling (Figure 1—figure supplement 2D). We also more strongly highlight the findings of the behavioral modelling, which indicate significant changes in animals’ usage of evidence and trial history in guiding decisions (subsection “A decision-making task for cerebellar investigations”).

From our data and modelling at this stage, we consider deficits at all stages of the decision-making process (ex. input sensory gain biases, integration impairments, post-categorization/premotor biases) possible and interesting to report. Thus, we do not claim that the cerebellum controls evidence integration per se. We have made this point clearer in the Results section and Discussion section.

Some of the shifts in psychometric curves may arise from asymmetry of injections (and we include specific evidence suggesting that such an asymmetry can account for some of the biases we see). We are currently doing experiments with newer technologies that we believe will resolve these questions in a more detailed manner.

However, we do note that the trend in our psychometrics is broadly consistent with effects seen in (Erlich et al., 2015), which shows that curves with this general phenotype (compression and bias) could conceivably be explained by a number of hypotheses ranging from sensory and premotor deficits to accumulation deficits. While it is beyond our current goal in this manuscript to perform this level of modelling, we use it just to demonstrate that our phenotype does not rule out interesting possibilities.

Drift diffusion models

Fits to drift diffusion models are an interesting direction for these data, yet also not necessary for our aim to demonstrate general causality for the region in the task. For completeness, we include below preliminary fits to a drift diffusion model showing what appears to be an increase in accumulation noise as a result of muscimol inactivation.

Given the poor statistical power of this dataset for this analysis, we don’t feel it substantially changes our main novel point that the cerebellum indeed plays causal role in some component of perceptual decision-making. We are currently performing a temporally specific inactivation study that will be better powered to perform this level of analysis. We intend to pursue more detailed modelling there.

**Author response image 1. respfig1:** Fits to a 5-parameter drift diffusion model (excluding the adaptation, initial noise, and sticky bounds found in the (Brunton, Botvinick and Brody, 2013) model). Best-fit parameters in the muscimol and baseline conditions are plotted on the likelihood landscape of the muscimol fit. The top panel indicates a tradeoff between sensory and accumulation noise in the fit to muscimol inactivation trials. However, the statistical confidence intervals for the muscimol fits span the displayed range for the lapse and accumulation noise parameters.

2) Evidence for evidence accumulation signals from imaging of somatic calcium in Purkinje cells: The authors report that appear similar to evidence-dependent and choice-selective ramping responses previously reported in many cortical and subcortical areas of primates and rodents. The manuscript reports temporal dynamics of Purkinje cell somatic calcium signals that, in many cells, manifest as a gradual ramping response during the cue period that is correlated with both the number of pulses on one or the other side (evidence) and/or the animal's eventual response (choice). These responses are interpreted with analogy to the ramping responses in trial-averaged spike rates recorded from parietal cortex (along with other forebrain regions) that are thought to encode the time-integral of sensory evidence. However, the manuscript does not convincingly demonstrate that cerebellar neurons are actually encoding accumulated evidence. In general, given the uncertainty about the origin of these signals, more caution is warranted in interpretation of the apparent ramping signals. Specifically:

We thank the reviewers for pointing out that our claims on this topic came across stronger than intended. As suggested, we have edited our interpretations to be more cautious. Our adjustments clarify what we now intend to claim and eliminate vagueness. We agree, as we note in the responses below, that (1) we have not demonstrated that cerebellar neurons precisely encode the stepwise accumulation of evidence, (2) the coding strategy within the cerebellum may be different than that of neocortex, and (3) our measurements are limited in the ability to answer (1) and (2). The analogy to more finely resolved neocortical signals is intriguing but not crucial to our claims, being more of a discussion point. We have therefore re-worded our conclusions to emphasize these distinctions.

On that note, some of the reviews in this section are concerned with how evidence is specifically represented, and how that may relate to specific evidence-accumulation models. As we note at the start of our response, an accumulation role per se for the cerebellum is possible but beyond our data to support. In this first contribution to the study of the cerebellum in this task, we focus on findings that precede such a claim. We consider our main finding to be that choice- and evidence-related information is present with sufficient fidelity to play a role in decision-making computations.

(Note that point 2a and our accompanying response is found below alongside point 2d).

b) Given the temporal resolution of the calcium indicator, it is difficult to interpret any evidence-related signal dynamics as reflecting an underlying ramp in neuronal firing rate. The original description of the use of somatic calcium imaging to track changes in Purkinje cell simple spike rate (Ramirez and Stell, 2016) found that these signals were so slow that step changes in simple spike rate could result in ramps of fluorescence like those seen here. This technical concern would best be addressed by at least some recordings comparing the calcium signals to simple spike rate with electrophysiology. Although the authors have tried to be careful in their writing, the text currently leaves plenty of room for misinterpretation by less technical readers. They should clarify the text further to be much more clear about the limitations of interpreting the temporal dynamics of the calcium signals.

We have clarified the text to emphasize the limitations in interpreting our calcium signals (subsection “Purkinje cell somatic calcium encodes task-relevant information”, Discussion section). We have more strongly noted that step changes in firing rate are indeed possibly the underlying representation.

While simultaneous calcium and electrical recordings are beyond our current technical abilities, we have obtained electrical recordings of Purkinje cells during performance of the task in trained animals. These recordings substantiate our primary findings of task-modulated signalling and suggest that the trial-averaged calcium signals we observe indeed correspond to ramps in firing rate. These recordings are now included in Figure 2—figure supplement 2. We include these important data, but nevertheless remain conservative in our interpretations of calcium data (Discussion section), especially since the nature of these signals remains unresolved at the single-trial level.

c) The relevant decision variable in the pulse accumulation task is the *difference* in pulses between the two sides. The authors seem to know this point well based on how they plot the psychometric functions. Yet at best a small minority of Purkinje cells encode this value. Subsection “Neuronal signatures of choice and evidence in Purkinje cells” reports that 39/843 cells (less than 5%) encode either a sum or difference of the number of pulses on each side. The specific figure for the number of cells encoding a difference is not reported, but Figure 3 suggests that it is about 5 cells (so ~2% of evidence-modulated cells or ~0.5% of all cells). These numbers question strong conclusions about the representation of the decision variable and accumulated evidence in cerebellum. Further, this result could be taken to show that responses in rodent cerebellum are different from those in primate cortex that are often interpreted as a representation of the decision variable for perceptual discrimination. Notably, it also indicates a dissociation between rodent cerebellum and neocortex, as Scott et al., estimate that as many as 1/3 of evidence-modulated cortical cells are better explained by the difference between sides. Therefore, the data strongly suggest that the critical decision-making operations are actually implemented downstream of the recorded cerebellar population. This strikes me as highly relevant to how the overall results should be interpreted, and it should be emphasized more strongly in the summary and discussion.

While it is true that a subset of cells in (Scott et al., 2017) represented the #R-#L quantity, a key conclusion of that study was that left and right evidence may be accumulated mostly independently. See their first discussion paragraph: “Behavioral Analysis and Neural Recordings Support the Existence of Two Weakly Coupled Accumulators.” In this light, we think that representations of single-sided evidence could possibly be important contributors to a decision-making circuit.

Nevertheless, we agree with the reviewer that cerebellar computations might not be the site of accumulation per se but possibly a filtering step preceding or following accumulation, as a node in a complex decision-making circuit. It may be the case, for example, that the cerebellar contribution is related to the individual accumulators proposed by (Scott et al., 2017), and that the weak coupling occurs in neocortex or elsewhere. We have now emphasized this point (Discussion section).

(We respond to the following two reviews, 2a and 2d, together below):

a) While there appear to be lots of time-modulated signals, the link to evidence accumulation and choice is not clear. It is apparent that the most salient feature of the recorded cerebellar population is a constant modulation, either upwards or downwards, with the passage of time in the trial. The presence of this signal complicates the interpretation of any ramping activity as being attributable to evidence accumulation. However, the statistical analyses (i.e. the linear model reported in subsection “Neuronal signatures of choice and evidence in Purkinje cells”) ignore this component of the responses. This might be valid if the representation of time were truly independent from the representation of evidence. But no such independence is established. For example, cells that are more strongly driven by evidence might also be more strongly driven by time, or more likely to be driven by time in one direction or another. The authors should more carefully consider the relationship between the different components of the cerebellar responses and/or formally control for potentially confounding effects of the time-related responses in the statistical models.d) The analysis of the representation of accumulated evidence (subsection “Neuronal signatures of choice and evidence in Purkinje cells” and Figure 3) ignores the time course of representation, focusing on a short window immediately before the decision. Even if we disregard technical problems explained above in 2a-c, it is unclear whether the cerebellar neurons represent integration of evidence over time or merely its final outcome. Note that the population analysis for the representation of evidence (Figure 2F) does not answer this question (and is generally not quite informative) because evidence is defined as the "correct" choice in that analysis rather than the magnitude of evidence (#R-#L).

Points 2a and 2d are each distinct and important points, which concern the temporal representations observed in Purkinje cells in our study. We address the specifics of these concerns below, but we note generally that their relevance is contingent on an assumption that we are claiming cerebellar neurons represent integration of evidence per se. While we are aware of studies that have made strong claims about the precise representations of accumulated evidence in single-neuron signals, we are not making claims as strong. We have modified the text in the manuscript to clarify this (subsection “Dynamics of choice- and evidence-related information in Purkinje cells”, Discussion section).

Our intention is to demonstrate that we observe evidence-related information in neuronal activity. Specifically, we make 2 points about evidence representation in our data:

(1) Stimulus information is present throughout the cue period: evidence-side decoding in Figure 2F demonstrates population-level encoding of evidence-related information throughout most of the cue period (which, as noted by the reviewer, is not necessarily of the #R-#L value, and we refer to review 2c where we have responded to this point).

(2) The magnitude of single-sided evidence is present in some neurons (via an unresolved temporal encoding scheme): shown by linear model for evidence strength effects in Figure 3C.

These analyses are comparable to the type shown in Figure 5 of (Scott et al., 2017), which demonstrates the presence of task-relevant information but does not resolve its time course. Further, based on new data given in response to review 4a, we know that these representations are task-specific.

Notably, we do notclaim that:

(1) cerebellar modulation is entirely driven by evidence. Some level of time modulation, as suggested in review 2a is likely in our view, though we note that our signals cannot be exclusively a time representation because of the analyses mentioned above.

(2) cerebellar signals represent the integration of evidence on a moment-to-moment basis. Our resolution and signal-to-noise does not allow us to answer this yet. Even if our neuronal population encodes evidence by, for example, a distributed temporal basis, the presence of the necessary information at all is an important finding.

Additionally, we know from our upcoming study of neural correlates in the primary and secondary visual cortices that even apparently simple sensory responses (time-locked to individual stimuli) are modulated by time and numerous other factors (Koay et al., 2018). The slow calcium time dynamics of Purkinje cell somata makes disentangling time and evidence effects very difficult, and this is better addressed by specific experiments e.g. with carefully designed sampling of stimulus times, which is beyond the scope of our current study.

In summary, we think that the concern in 2a (evidence and time modulations may be intertwined) and the concern in 2d (neuronal signals may not directly track evidence over time) are interesting questions we have not resolved, but that our core conclusions do not depend on the answers. Finally, we note that we are currently performing electrical-recording experiments that address this important issue as part of a follow-up study.

3) "Error-related" dendritic responsesIn addition to somatic calcium signals, the authors analyzed dendritic calcium signals and found that these were higher on error vs. correct trials. The implications of this observation are emphasized heavily; for example, the Discussion section concludes that "the cerebellum, which learns from error to guide action, may help in the learning and tuning of accurate responses". This is an interesting proposal, but one might have had the same belief prior to seeing the results reported here (given existing human data and theories about the cerebellum), and it's not clear how the data should update it.

In this work we focused on reporting what is, to our knowledge, the first observation of error-related signalling in the Purkinje cell dendritic pathway during a cognitive, decision-making context. We agree with the reviewers that we did not resolve many interesting details of this signalling, in some part due to limitations of the experiment as explained in the specific replies below. Nevertheless, we feel that our result builds a bridge between known roles of error signals in motor contexts and hypothesized but never-observed roles in the domain of decision-making.

Beyond describing the finding, at this time we feel limited in our ability to propose specific roles of the observed dendritic signalling. We have therefore removed the heavy emphasis (like that referenced by the reviewer above). We still consider these findings an important result to include as they motivate future studies that we are currently pursuing, and we hope that they will also be of use to others given the current interest in this topic (Sendhilnathan et al., 2018; Wagner et al., 2017).

a) There is speculation about how the observed error responses could be used as error signals to guide learning, in line with existing models of cerebellar learning. However, it was not clear to the reviewers how that would work, especially given that the error responses observed were not directional. Typically, Purkinje cell complex spikes are thought to provide a directional signal for learning, not just a correct/ incorrect signal. There are no analyses to support the proposed functional role of these error related responses as being involved in "tuning" responses or correcting errors. Yet, it should be possible to provide a more thorough analysis of the error signals:What do error signals predict about future behavior? An obvious analysis would be to ask whether the magnitude of the error signal on trial n−1 influences choice accuracy on trial n. This seems to be a clear prediction from the proposal that the cerebellum "tunes" behavior or "corrects errors".

We appreciate the trial-by-trial analysis suggestion and we are very interested in knowing the answer to this question as well, especially given the success of similar analyses in recent cerebellar work on head movements (Brooks, Carriot and Cullen, 2015), eye movements (Medina and Lisberger, 2008), and eyeblink conditioning (Ten Brinke et al., 2017). We attempted this analysis, and encountered an issue of statistical power that is related to slow learning in this behavior and therefore prohibitively low trial counts.

To illustrate the point, we have performed a power analysis considering the number of correct and error trials in our dataset. To detect a 10% difference from mean performance (at α=0.05) in fraction correct choices, one needs approximately 16x the number of trials we collected in our imaging study. We also note that this is under a very generous assumption that there would be a 10% difference in performance as a result of dendritic signalling within a session, but we know that in our task, small improvements in performance take place over weeks (learning occurs on the order of thousands of trials with gradually diminishing error rates). This is in contrast to the behaviors used in studies that have successfully performed such analyses, where learning is substantially faster and can thus be expected to be accompanied by robust neural correlates in just tens of trials.

We are encouraged to have found this elevated signalling and now that we know it exists, it may be possible to design a study to test this, but it would best involve a new or modified experimental design. We have added text to the Discussion on this topic.

c) How do the error signals relate to the strength of evidence on each trial? This is briefly mentioned at the end of the Results section and in Figure 4—figure supplement 1, but without any statistical tests or interpretation. Whether and how the error signals are modulated by the strength of evidence seems key to determining their functional role in updating behavior, if any. In particular, we would have expected that true "error" responses would be higher for easier trials, but the opposite appears to be true.

We have added statistics to the reporting of these results, which proved to be not significant despite a possibly interesting trend. If the trend had been significant, it may have been compatible with inverted reward signalling which is known to occur in some other brain regions (Cohen, 2007; Matsumoto and Hikosaka, 2007). We now comment on this in the manuscript (Discussion section).

4) Ruling out sensory and motor confounds as potential sources of somatic and dendritic calcium signalsa) Figure 3B indicates that isolated puffs produce a calcium response that gradually rises and falls over ~1 s. On trials with strong evidence, these signals will overlap. Assuming that they sum reasonably linearly, overlapping calcium responses from neurons that encode only the transient presentation of evidence would nevertheless give the impression of a gradual ramp with a slope that depends on the quantity of evidence. Note that this is a different issue from the point raised in the Discussion section about distinguishing single-trial ramps from steps. Rather, it makes it unclear whether the evidence-related cerebellar responses correspond to a representation of the momentary sensory evidence or to the magnitude of accumulated evidence that drives choice. This distinction is critical to interpreting proposed neural implementations of evidence accumulation models. One possible way to address this would be to record responses during stimulus presentation from animals that are not engaged in making a decision.

We see the importance of the suggested control, and we have included new data from mice subjected to trials under identical conditions but not trained in the decision-making behavior. These data can be seen in Figure 3—figure supplement 1 (and subsection “Dynamics of choice- and evidence-related information in Purkinje cells”), and they indicate an absence of ramping, evidence-side decoding, or puff count representation in these signals. We cite this, and the electrical recordings given in response to review 2b as evidence that the calcium modulation we observe is not a result of simple sensory responses.

b) The authors attempt to rule out that the somatic signals are related directly to movement of the animal and less related to the "cognitive" variables. The analysis in the supporting figures is in line with the level of detail often applied in the field currently. But, is it possible that the activity is related to other movements of the animal rather than in the orofacial region? The authors should either cite strong evidence showing that these are the only relevant regions for the area of cerebellum examined or provide additional videography results from other parts of the body (e.g. paws). This is of course a significant point to make as strong as possible given the cerebellum's long-established role in motor behaviors.

We now cite evidence of primarily orofacial representations in crus I (subsection “Purkinje cell somatic calcium encodes task-relevant information”). In particular, we note here that “in rats, the largest whisker projection area in the cerebellar cortex is located in crus 1, occupying the largest part of that folium” (Bosman et al., 2010) (and crus 1 features are well conserved in rodents (Sugihara, 2018)), and that a comprehensive review of cerebellar representations concludes “the physiological role of crus I and crus II in controlling forelimb and hindlimb muscles could no longer be maintained” (Manni and Petrosini, 2004).

Nevertheless, we now include more movie-based measurements of forepaws, which also support the conclusion that movements do not account for the somatic signals (subsection “Purkinje cell somatic calcium encodes task-relevant information”, Figure 2—figure supplement 4, Video 3).

We appreciate the reviewers’ concerns and do not believe we can extensively rule out all movements in the body (including other body parts not visible, and other more subtle movements). That said, it is a less likely explanation than the one we propose, given the known roles of this region and our inactivation results. We have demonstrated that the strongest expected movement-related signals, orofacial movements, are not responsible for the somatic activity that we observe. By adding forepaw analyses, we have furthermore shown that less-likely movement-related signals are also not an explanation. This standard of proof is comparable that rest of the field of evidence accumulation.

c) The analyses in Figure 4—figure supplement 1 do not convincingly rule out the possibility that the difference in dendritic calcium signals on error trials resulted from differences in licking on error trials. That figure clearly shows that cessation of licking proceeds at different rates on error trials compared to other times. How does trial type (correct vs error) affect the relationship between licking (or changes of rate in licking) and dendritic calcium signals?

We take this concern very seriously and we have thought hard about what analysis properly addresses it. We believe the reviewer’s question is addressed by our analyses in Figure 4F and Figure 4—figure supplement 1. We now explain our rationale more thoroughly in the text (subsection “Error-associated signalling in Purkinje cell dendrites”), as well as include an additional panel in Figure 4F to visually demonstrate the rationale of the analysis.

To our understanding, the specific null hypothesis we need to reject is that the elevation in dendritic signalling with errors is reflective not of an error event but rather of a specific motor event that tends to co-occur with errors. The most likely such motor event would be the cessation of licking, since this is the one that reliably occurs at the moment of errors (as the reviewer notes that we show in Figure 4—figure supplement 1A), but others, such as initiationof licking (perhaps toward the other direction) or movements of other sorts are possible too.

The reasoning of our analysis is as follows: consider first the hypothesis that dendritic activity increases not specifically with errors but rather when animals stop licking, an event that nearly always occurs when animals make errors. If this were true, then the cessation of licking, whether or not it occurs at the moment of error, would elicit this elevated signalling. We take advantage of the fact that we have many additional instances where lick cessation occurred *not* at the moment of errors. These “non-error” lick cessations occur in all correct trials, as noted by the reviewer. We computed the dendritic responses at all such moments of lick cessation, in both error contexts and non-error contexts (i.e. correct trials). We then compared the responses on a cell-by-cell basis across these two contexts, yielding an error:correct activity ratio. If the best explanation for the elevated signalling were a lick cessation-related signal, then these lick cessation events should elicit responses of similar magnitude (i.e. error:correct activity ratio of 1), but we find a statistically elevated level of signalling specifically when lick cessation occurs with errors, as compared to when it occurs elsewhere. This implies that signalling is specifically elevated at moments of error beyond what is expected just when animals stop licking in non-error contexts. In other words, lick cessation in error trials elicits more dendritic response than lick cessation in correct trials–a direct response to the question “How does trial type (correct vs error) affect the relationship between licking and dendritic calcium signals?”

We apply the same logic to various definable motor events, such as moments of low, medium, and high rates of licking, as well as lick initiation and orofacial movements. We also apply the lick initiation and cessation analyses over differing time windows, including the dendritic activity preceding and following the lick initiation/cessation, to account for the possibility that a lick-related signal in the cerebellum temporally precedes or follows the licking measurement.

We consider this to be the most thorough approach we can take to answer whether there exists elevated error-related dendritic signalling beyond what is expected from specific motor hypotheses.

To clarify this analysis for readers, we have added an additional panel (in Figure 4F) illustrating this type of measurement, and relabelled the descriptions to align with the “correct vs error” phrasing given by the reviewer (Figure 4F, Figure 4—figure supplement 1B).

[Editors' note: further revisions were requested prior to acceptance, as described below.]

The manuscript has been improved but there are some remaining issues that need to be addressed before acceptance, as outlined below:The authors have made a concerted effort to clarify the text of their paper and have provided reasonable responses to many of the points raised. As was noted in the first round of reviews, rebuttal, and in the text, it remains unclear what the cerebellum's specific role or computation is during the task, and the reviewers were somewhat disappointed that most of the key questions are deferred to future studies. However, the reviewers also felt that defining cerebellum's exact role in the task may be asking too much for a first study, particularly given the limitations of the dataset and the conceptual and analytical complexities that prevent the authors from specifying the nature of neural representations and the source of the behavioral deficit following cerebellar inactivation.Overall, there are several interesting leads about the role(s) of cerebellum in perceptual decisions in this paper. While none of these are firmly established by the current study, the reviewers agree that the writing and presentation of the results is generally fair and not greatly over-stated, and favor publication so that the results can be evaluated by the field and followed up in future studies from other groups.The reviewers note that in the process of clarifying what can and cannot be claimed based on the existing data, the scope of the paper is limited to three points: successful training of mice to do the task, reduced accuracy following cerebellar inactivation, and representation of task-relevant variables in cerebellar neural population without specifying the exact nature of the represented variables. Before publication, the reviewers agree that it is important to ensure that after toning down its claims, the manuscript does not leave behind any statements that could mislead readers as to what is actually shown. The reviewers have identified the following statements from the Title, Abstract, and Discussion section that are potentially misleading and should be restated with more specific statements that more accurately match the conclusions that can be supported by the data.TitleA cerebellar role in evidence-guided decision-making.

Cerebellar involvement in an evidence-accumulation decision-making task.

Impact StatementThe lateral posterior cerebellum participates in evidence-accumulation-based decision-making, and Purkinje neurons in this region encode choice-, evidence-, and error-related variables. [Suggest replacing this stronger statement with language more like that used in the rebuttal, such as "choice- and evidence-related information is present in lateral posterior cerebellum and could participate in decision-making computations during a decision-making task involving evidence-accumulation."].

In a new evidence-accumulation decision-making task, activity of the lateral posterior cerebellum is necessary for accurate performance, and Purkinje cell somatic and dendritic activity contain choice/evidence and error-related information.

Abstract- Here we show that during perceptual decision-making over a period of seconds, decision-, sensory-, and error-related information converge on the lateral posterior cerebellum in crus I, [The presence of task-related signals is shown. Convergence is not, and decision and sensory signals are not clearly dissociated].

Sentence removed from abstract.

- Demonstrated that cerebellar inactivation reduces behavioral accuracy without impairing motor parameters of action [Not all motor parameters were controlled for].

(Abstract) Cerebellar inactivation led to a reduction in the fraction of correct trials.

- We found that Purkinje cell somatic activity encoded choice- and evidence-related variables [Please avoid the suggestion that the specific variables that are encoded have been determined].

(Abstract) “.…we found that Purkinje cell somatic activity contained choice/evidence-related information”.

- Decision errors were represented by dendritic calcium spikes, which are known to drive plasticity [This could misleadingly suggest that they are known to drive plasticity in this context].

(Abstract) “Decision errors were represented by dendritic calcium spikes, which in other contexts are known to drive cerebellar plasticity.”

- We propose that cerebellar circuitry may contribute to the set of distributed computations in the brain that support accurate perceptual decision-making. [Should be more focused on task performance].

(Abstract) “We propose that cerebellar circuitry may contribute to computations that support accurate performance in this perceptual decision-making task."

Discussion section- Cerebellar inactivation reduces animals' use of evidence and increases their use of choice history. [given the limitations of the interpretation of the inactivation experiments, this statement should be more conservative].

(Discussion section) Our fits to a behavioral choice model indicate that reduced performance was accompanied by a decreased weighting of evidence and increased weighting of choice history parameters.

- Given the temporal resolution of calcium measurements, our somatic signals may correspond to firing rate ramps (Shadlen and Newsome, 2001), steps (Latimer et al., 2015), or more complex response profiles that form a temporal basis for evidence accumulation (Scott et al., 2017). [This statement, as well as the corresponding section of the Results section, should include an explicit reference to the time course of somatic calcium signals from Purkinje cells, which is at least an order of magnitude slower than typical calcium imaging (Ramirez and Stell, 2016).].

(Discussion section) Cytoplasmic calcium acts as a temporally filtered readout of firing rate, limited by calcium removal times that are slower in Purkinje cells (Konnerth et al., 1992; Lev-Ram et al., 1992; Fierro and Llano, 1996; Rokni and Yarom, 2009; Ramirez and Stell, 2016) than in neocortical neurons (Chen et al., 2013).

(Subsection “Purkinje cell somatic calcium encodes task-relevant information”) Cytoplasmic calcium acts as a temporally filtered readout of firing rate, and calcium extrusion in Purkinje cells occurs on a slower time scale (Konnerth et al., 1992; Lev-Ram et al., 1992; Fierro and Llano, 1996; Rokni and Yarom, 2009; Ramirez and Stell, 2016) than in neocortical neurons (Chen et al., 2013). Therefore, our observed increasing and decreasing time courses of calcium could reflect various firing rate profiles, such as impulse responses, ramps, or steps.

-The task-modulated activity we observe encodes both choice-related and evidence-related variables that may be used during the decision-making process. [Please avoid the suggestion that the specific variables that are encoded have been determined].

(Discussion section).…the neural activity we observed contains task-relevant information that may be used during evidence accumulation and decision-making.

- We observed an excess of dendritic calcium events coincident with decision errors, demonstrating for the first time observations compatible with error-associated signalling in a decision-making reward context. [given the limitations of the interpretation of these signals, this statement should be more conservative].

(Discussion section) We observed an excess of dendritic calcium events coincident with decision errors, which has not been previously reported in the cerebellum.

[Editors' note: further revisions were requested prior to acceptance, as described below.]

The manuscript has been improved but there are some final issues that need to be addressed before acceptance, as outlined below:In response to the request to re-evaluate the evidence for ramping signals that could be obtained with the somatic calcium imaging, the authors now state (Subsection “Purkinje cell somatic calcium encodes task-relevant information”), "Therefore our observed increasing and decreasing time courses of calcium could reflect various firing rate profiles, such as impulse responses, ramps, or steps."In light of this revision, as well as the fact that the electrophysiological evidence provided in Figure 2—figure supplement 2 is from only a few cells, all of which show positive ramps of activity, the following statements should also be revised:- (Subsection “Purkinje cell somatic calcium encodes task-relevant information”), "We did find that electrically recorded Purkinje cells exhibited gradually increasing rates of firing throughout the cue period (Figure 2—figure supplement 2), suggesting that on average across trials, the fluorescence signals we observed correspond to firing rate ramps."

(Subsection “Purkinje cell somatic calcium encodes task-relevant information”) “We did find that electrically recorded Purkinje cells exhibited gradually increasing rates of firing throughout the cue period (Figure 2—figure supplement 2).”

- (Discussion section), "Our electrical recordings also showed ramps, suggesting that temporally filtered firing rate ramps are sufficient to account for our observed fluorescence signals."

(Discussion section) “Our electrical recordings also showed ramps, consistent with the idea that temporally filtered firing rate ramps are sufficient to account for our observed fluorescence signals.”

The reason we went to the effort to obtain and include electrical recordings, per suggestions of many of our colleagues, was that it could help substantiate the possibility of fluorescence ramps corresponding to firing rate ramps. We think that the Discussion is an appropriate place to at least note the relationship between these two findings and our interpretation of them.

[Editors' note: further revisions were requested prior to acceptance, as described below.]

Your revised article has been favorably evaluated by our editors again, but there remain some issues that need to be addressed before acceptance, as outlined below. Given that this is the third request for revisions, we will be unable to follow with any more. Please attend to this final issue one way or the other so that the next letter will be the final one.We appreciate the authors' desire to speculate here. However, in our view, the "suggests/ consistent with" was not the only problem with this sentence. There is also a problem with "sufficient". The electrophysiological evidence in Figure 2—figure supplement 2 is anecdotal and non-quantitative. For this statement to be left in, it would need to be adequately supported. In our view, this would require a quantitative comparison between imaging and electrophysiology results. In particular, we would want to know:- How many Purkinje cells in total were recorded from electrophysiologically? How many of these showed ramping? (all of the cells they showed us show positive ramps, but it is not clear if those were selected from a larger data set)- Did any Purkinje cells show ramping calcium signals without a transient increase in firing rate?- Did any Purkinje cells show ramping calcium signals without ramps in firing rate (for instance, in cases where only a transient increase in firing may have been observed electrophysiologically)?- Why are no decreasing activity ramps found with electrophysiology, but they are found with imaging?- What accounts for the decreasing ramps that were observed with calcium imaging?- What would the predicted calcium signals be for the examples shown if the spike rates recorded electrophysiologically (with and without the transient increase/ ramping components) were convolved according to Ramirez and Stell (2016)? And/or with the authors' own convolution/ deconvolution methods, from the simultaneous calcium imaging/ electrophysiological recordings that they performed?We give the authors the choice of either fully addressing these points, or using compromise language, such as "Preliminary electrical recordings also showed ramps, consistent with the idea that temporally filtered firing rate ramps may account for the observed fluorescence signals."

We thank the reviewers and editors for their suggestions. We have made the requested adjustment to the manuscript text, opting to use the exact compromise phrasing beginning with “Preliminary …” proposed by the editors. Below we include the original sentence for reference, followed by the updated sentence:

(Discussion section), “Our electrical recordings also showed ramps, consistent with the idea that temporally filtered firing rate ramps are sufficient to account for our observed fluorescence signals.”

(Discussion section) “Preliminary electrical recordings also showed ramps, consistent with the idea that temporally filtered firing rate ramps may account for the observed fluorescence signals.”